# Improving Continual Learning Performance and Efficiency with Auxiliary Classifiers

Filip Szatkowski [1 2]   Yaoyue Zheng [3 4]   Fei Yang [5 6]
Tomasz Trzciński [1 7 8]   Bartłomiej Twardowski [7 4]   Joost van de Weijer [4 9]

## Abstract

Continual learning is crucial for applying machine learning in challenging, dynamic, and often resource-constrained environments. However, catastrophic forgetting — overwriting previously learned knowledge when new information is acquired — remains a major challenge. In this work, we examine the intermediate representations in neural network layers during continual learning and find that such representations are less prone to forgetting, highlighting their potential to accelerate computation. Motivated by these findings, we propose to use auxiliary classifiers (ACs) to enhance performance and demonstrate that integrating ACs into various continual learning methods consistently improves accuracy across diverse evaluation settings, yielding an average 10% relative gain. We also leverage the ACs to reduce the average cost of the inference by 10-60% without compromising accuracy, enabling the model to return the predictions before computing all the layers. Our approach provides a scalable and efficient solution for continual learning.

## 1. Introduction

The ability to adapt to changing environments is crucial for practical machine learning applications, particularly in resource-constrained settings where computational efficiency is essential. Continual learning provides the theoretical foundations and algorithms for learning from non-i.i.d. data streams in such environments (De Lange et al., 2021).

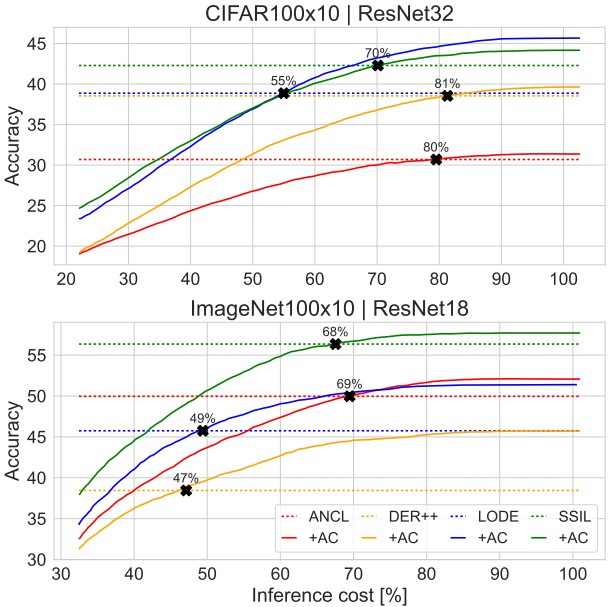

*Figure 1.* We integrate auxiliary classifiers (ACs) into various CL methods, enabling dynamic inference and reducing the inference cost. We measure their accuracy relative to the standard network at different computational budgets and show that AC-enhanced methods match the performance of the standard counterparts at only 50-80% cost, and improve their performance at higher computational budgets. The accuracy of AC-enhanced models saturates at 80-90% computation, allowing us to save 10-20% of the inference cost without sacrificing accuracy.

The most common continual learning scenario involves learning from sequential tasks, where the learner cannot access previously seen tasks when learning new ones. The tasks may differ in data distributions (domain-incremental learning) or introduce new classes (class-incremental learning) and may or may not provide task identity during classification (task-incremental learning) (Van de Ven & Tolias, 2019). The primary challenge in continual learning is to avoid *catastrophic forgetting* – a significant drop in performance on past tasks while learning on the new data (McCloskey & Cohen, 1989; Kirkpatrick et al., 2017). Various strategies including parameter isolation (Rusu et al., 2016; Serra et al., 2018; Mallya & Lazebnik, 2018), weight

[1]Warsaw University of Technology [2]IDEAS NCBR [3]Institute of Artificial Intelligence and Robotics, Xi'an Jiaotong University, China [4]Computer Vision Center, Barcelona [5]VCIP, College of Computer Science, Nankai University [6]NKIARI, Shenzhen Futian [7]IDEAS Research Institute [8]Tooploox [9]Universitat Autonoma de Barcelona. Correspondence to: Fei Yang <feiyang@nankai.edu.cn>.

*Proceedings of the 42nd International Conference on Machine Learning*, Vancouver, Canada. PMLR 267, 2025. Copyright 2025 by the author(s).

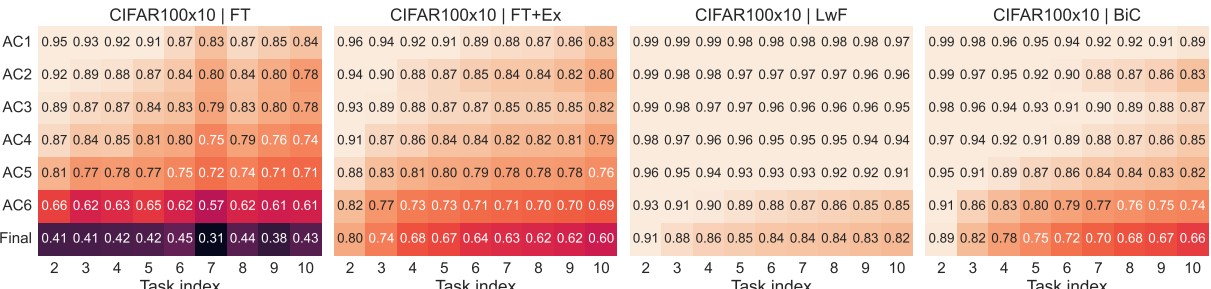

*Figure 2.* CKA of the first task representations across different ResNet32 layers (L1.B3-L3.B5) through continual learning on CIFAR100 split into 10 tasks. Representations at the early layers are more similar across the continual learning, hinting at the potential for more stability that could be leveraged to improve the performance by incorporating auxiliary classifiers at the intermediate layers.

and data regularization (Kirkpatrick et al., 2017; Li & Hoiem, 2017), and rehearsal methods (Rebuffi et al., 2017; Chaudhry et al., 2018) have been proposed to address this challenge. Despite these efforts, continual learning, particularly class-incremental learning, remains an open problem.

Several studies have investigated knowledge transfer and accumulation in deep neural networks through intermediate representations (Evci et al., 2022; Jung et al., 2023), and works such as Liu et al. (2020a); Ramasesh et al. (2020); Zhao et al. (2023); Masarczyk et al. (2023) revealed that later layers undergo the most significant changes during continual learning. However, these insights have yet to be leveraged to enhance continual learning performance. Independently, work on neural network efficiency has introduced early-exit classifiers (Panda et al., 2016; Teerapittayanon et al., 2016), which utilize intermediate layers to make predictions before reaching the final classifier. Such classifiers not only accelerate inference but can also improve accuracy by addressing "overthinking" – a phenomenon where inputs that could be correctly classified earlier are unnecessarily processed by deeper layers, sometimes leading to errors (Kaya et al., 2019). In this paper, we bridge these two research areas and propose to use auxiliary classifiers (ACs) in continual learning to capitalize on the lower forgetting observed in early layers and enable computational savings during inference.

We begin our work with a detailed analysis of intermediate representations in continual learning. First, we examine the stability of representations across different network layers using CKA (Figure 2) and find that early layer representations are more stable throughout the training. Building on this insight, we explore the discriminative power of these representations and reveal that, surprisingly, auxiliary classifiers (ACs) trained on top of the early layers can significantly outperform the final classifier on older tasks. To dive deeper into this phenomenon, we compare overthinking in continually trained networks with models trained in i.i.d. settings, discovering that overthinking is far more pronounced in continual learning. Our findings highlight the potential of

intermediate representations in continual learning scenarios, suggesting that utilizing additional classifiers built on top of these representations could effectively reduce forgetting.

Motivated by the insights from our analysis, we adapt and integrate auxiliary classifiers (ACs) into various well-established continual learning strategies (LwF (Li & Hoiem, 2017), EWC (Kirkpatrick et al., 2017), ER (Chaudhry et al., 2019), BiC (Wu et al., 2019), SSIL (Ahn et al., 2021), ANCL (Kim et al., 2023), LODE (Liang & Li, 2024), DER++ (Buzzega et al., 2020)) and demonstrate that they consistently outperform single-classifier methods on standard benchmarks such as CIFAR100 and ImageNet100. Since ACs enable dynamic inference and control of the network computation through early exit, we also explore the efficiency gains offered by their introduction. As shown in Figure 1, AC-based networks with dynamic inference can maintain the performance of single-classifier models while using only a fraction of the computation. Furthermore, at 80-90% of the original network's computational cost, performance reaches a saturation point, which means that our method provides lossless acceleration. Our results demonstrate that ACs can be seamlessly integrated into common continual learning methods, yielding consistent performance improvements and computational savings without the need for extensive hyperparameter tuning.

ACs provide a high-performing alternative to standard methods in scenarios where faster inference or flexible resource usage is essential. Our contributions can be summarized as:

- We analyze intermediate representations in continual learning (CL) and show that classifiers built on such representations are less prone to forgetting.

- We propose to use *auxiliary classifiers* (ACs) in CL and show that our idea yields an average 10% relative improvement over single classifier alternatives across diverse benchmarks and architectures.

- We leverage ACs to reduce the average cost of inference by 10-60% without sacrificing accuracy by allowing early prediction through *dynamic inference*.

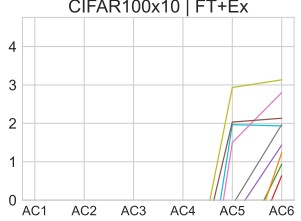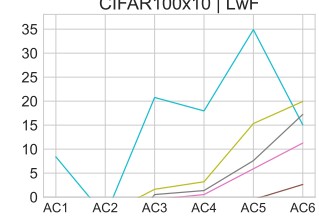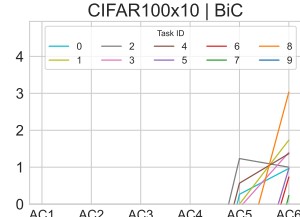

*Figure 3.* Per-task **difference** (only positives) in accuracy between the auxiliary classifiers (ACs), trained with linear probing on intermediate layers, and the final classifier. Surprisingly, in continual learning, some intermediate classifiers can significantly outperform the final classifier on the old task data, especially for exemplar-free methods (FT and LwF).

## 2. Related works

**Continual learning.** Continual learning methods (Parisi et al., 2019; De Lange et al., 2021; Masana et al., 2022) can be broadly categorized into three types: *regularization-based*, *replay-based*, and *parameter-isolation* methods. *Regularization-based* methods typically introduce a regularization term to the loss function to constrain parameter changes for prior tasks, with subcategories of data-focused (Li & Hoiem, 2017; Kim et al., 2023) and prior-focused (Kirkpatrick et al., 2017; Zenke et al., 2017; Aljundi et al., 2018) approaches. Recent research also enforces weight updates within the null space of feature covariance (Wang et al., 2021; Tang et al., 2021). *Replay-based* methods use memory and rehearsal mechanisms to recall past tasks during training, maintaining low loss on those tasks. Two main strategies are exemplar replay - which stores selected training samples (Riemer et al., 2018; Buzzega et al., 2020; Chaudhry et al., 2018; Prabhu et al., 2020; Chaudhry et al., 2019; Liang & Li, 2024), and generative replay, where models synthesize previous data using generative models (Shin et al., 2017; Wu et al., 2018). *Parameter isolation* methods learn task-specific sub-networks within a shared network. Techniques like Piggyback (Mallya et al., 2018), PackNet (Mallya & Lazebnik, 2018), SupSup (Wortsman et al., 2020), HAT (Serra et al., 2018), and Progressive Neural Networks (Rusu et al., 2016) allocate and combine parameters for individual task. While effective in task-aware settings, these methods are most suited for scenarios with a known task sequence or oracle.

**Representations in neural networks.** Understanding and comparing representations (Kornblith et al., 2019; Davari et al., 2022) at different layers in deep neural networks is an active area of research on transfer learning (Boschini et al., 2022) and continual learning (Ramasesh et al., 2020; Zhao et al., 2023; Masarczyk et al., 2023). Motivated by such analyses, several continual learning methods leverage intermediate layers for replay (Liu et al., 2020a; Pawlak et al., 2022) or regularization (Douillard et al., 2020). Early-exit techniques (Panda et al., 2016; Teerapittayanon et al.,

2016; Kaya et al., 2019; Wójcik et al., 2023) that attach intermediate classifiers to the model were developed based on the observation that their representations could be used for classification to allow skipping later model layers and reduce inference cost. While works such as Liu et al. (2020b); Yan et al. (2024) use multiple classifiers to improve online CL through specialized techniques such as ensembling, self-distillation, and contrastive learning, our approach explores intermediate classifiers as a general framework that enhances offline class-incremental learning performance and efficiency and can be applied to most continual learning methods and architectures.

## 3. Intermediate representations in CL

In this section, we analyze the stability of intermediate representations in continual learning and how *auxiliary classifiers* (ACs) trained on these representations can reduce forgetting. We train a neural network over the 10 tasks of split CIFAR100 (Krizhevsky, 2009) in a class-incremental learning setting (De Lange et al., 2021; Masana et al., 2022), aiming to classify new classes while avoiding catastrophic forgetting of previous ones *without* access to task identity at prediction time. For the analysis we train ResNet32 (He et al., 2016) with different approaches for continual learning: naive finetuning (FT) without any additional continual learning technique, finetuning with exemplars (FT+Ex), exemplar-free regularization with LwF (Li & Hoiem, 2017), and BiC (Wu et al., 2019) with both regularization, exemplars and an additional bias correction. We analyze representations at 6 intermediate layers and the final feature layer before the classifier, referring to the intermediate classifiers as AC1-AC6. See Appendix A for setup details and Appendix F.3 for a detailed description of our approach, further analysis, and more results.

### 3.1. Intermediate representations are more stable

We present the representational similarity of the intermediate and final layers in Figure 2 using Centered Kernel Alignment (CKA) (Kornblith et al., 2019). CKA measures

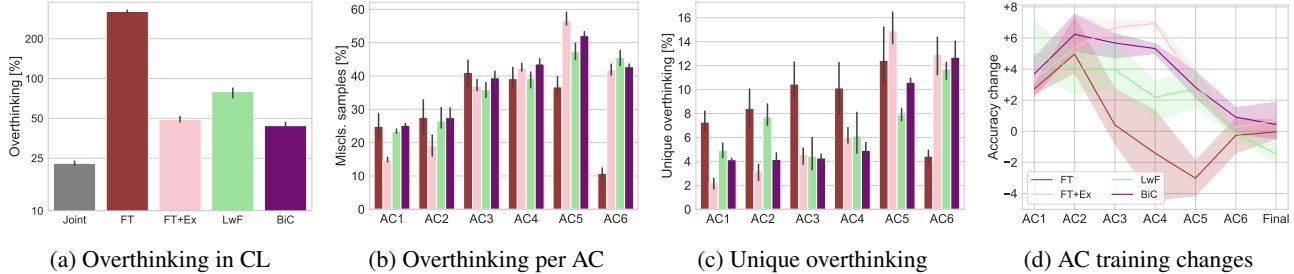

(a) Overthinking in CL     (b) Overthinking per AC     (c) Unique overthinking     (d) AC training changes

*Figure 4.* Overthinking and AC performance analysis on CIFAR100x10. Overthinking refers to a case where samples correctly classified by early classifiers are misclassified later by the final classifier. (a) Overthinking is much more prominent in continual learning methods than in standard joint training, which indicates that the accuracy of continual learning could be greatly improved through ACs. (b) Each classifier correctly classifies a significant portion of the samples misclassified by the final classifier. (c) Subsets of samples can be correctly classified **only** by a single given AC. (d) Training ACs together with final networks improves the performance of most classifiers.

the similarity of two sets of representations by computing the normalized Hilbert-Schmidt Independence Criterion (HSIC) between their kernel. It is invariant to orthogonal transformations and isotropic scaling, making it suitable for comparing learned features across layers or models. We measure the similarity between the representations of the first task data learned after the initial task and the representations learned after each subsequent task. Across all methods, early layer representations change less and exhibit more stability. As expected, the most significant changes occur in FT. Continual learning methods, such as FT+Ex, LwF, or BiC, show less change due to replay or regularization strategies that enforce stability, though the trend of early-layer stability remains. Our findings are consistent with the observations in previous research (Ramasesh et al., 2020; Zhao et al., 2023) and suggest that intermediate representations are more robust in continual learning, and classifiers built on top of those representations should suffer from less forgetting than the final classifier.

### 3.2. Early layer classifiers perform better on old data

Knowing that early layers produce more stable representations in class-incremental learning, we now assess their usefulness for classification. To assess the discriminative power of intermediate representations, we use linear probing (Davari et al., 2022). Specifically, we attach ACs to the intermediate representations of the network and train them alongside the main model in a continual scenario. To prevent interference with the representations during the learning of the ACs, we detach the gradients between the ACs and the rest of the network. Given the high dimensionality of the representations, the ACs consist of a simple pooling layer followed by a linear layer. Then, we compare the final accuracy of the learned ACs with that of the final classifier on a per-task basis and show the differences in Figure 3.

ACs perform surprisingly well, with the penultimate classifier's accuracy matching or surpassing the final classifier on

older tasks across all methods – an outcome not typically seen in i.i.d. settings. The most interesting results can be observed for the exemplar-free methods: in FT, intermediate classifiers achieve the highest accuracy for most of the tasks, and in LwF, the intermediate classifiers outperform the final classifier by a significant margin on the older tasks. For exemplar-based methods, deeper ACs also surpass the final classifier, though the differences are less prominent, and earlier classifiers do not exceed the final one as in the exemplar-free cases. Generally, early classifiers exhibit more stability, while later classifiers provide better ability to discriminate between classes. Our results suggest that selecting the appropriate AC could improve performance in continual learning, especially on older tasks.

### 3.3. Continually trained networks *overthink* more

*Overthinking* (Kaya et al., 2019) refers to cases where intermediate classifiers correctly classify samples that the final classifier misclassifies. This concept emphasizes that using classifiers that operate on intermediate representations can not only accelerate inference but also enhance network accuracy. To formally define overthinking, we consider a multi-classifier network evaluated under an oracle prediction rule, where a sample is considered correctly classified if any of the intermediate or final classifiers predicts the correct label. We denote the accuracy under this rule as $Acc_{oracle}$, and let $Acc_{final}$ represent the accuracy when using only the prediction of the last classifier. Overthinking is then measured as the performance gap: $O = Acc_{oracle} - Acc_{final}$, and we also define relative overthinking as the normalized quantity $O/Acc_{final}$. Initially, overthinking was studied in intermediate classifiers under i.i.d. settings. However, our prior analysis indicates that a similar effect occurs in continual learning, particularly for older tasks, where auxiliary classifiers (ACs) experience less forgetting than the final classifier. Therefore, we investigate overthinking in continual learning methods and compare it to the standard

i.i.d. joint training scenario in Figures 4a to 4c.

Networks trained using different continual learning methods exhibit a higher degree of overthinking compared to those trained under standard joint training (i.i.d. setting). To highlight the difference, we plot relative overthinking in Figure 4a. The more pronounced overthinking in continual learning aligns with our earlier findings, where intermediate classifiers perform surprisingly well and often outperform the final classifier on older tasks. In Figure 4b, we show the proportion of samples misclassified by the final layer that each AC correctly classifies, that is how much a given classifier contributes to overthinking. While the later layers correctly classify larger subsets of the samples, early layers still can contribute towards improving the performance, especially in exemplar-free methods. Since the correctly classified subsets might overlap between the classifiers, in Figure 4c, we further investigate how diverse the learned classifiers are by measuring unique overthinking. Specifically, we measure which samples, misclassified by the final layer, are correctly classified *only* by a specific AC, which emphasizes the diversity of the learned classifiers. The heightened overthinking in continual learning suggests that ACs preserve valuable knowledge that the final classifier struggles to retain. Unlike in the i.i.d. joint training case, where ACs primarily serve to speed up inference, our analysis suggests that in continual learning, they could also play a crucial role in improving overall performance.

### 3.4. End-to-end training improves ACs

Although ACs trained through linear probing show promise for improving performance in continual learning, it is unclear how end-to-end training with enabled gradient propagation would affect the network. To explore this, we train the network with gradient propagation and compare the final average accuracy across all tasks with linear probing classifiers. We show the difference in the classifiers' performance in Figure 4d. End-to-end training improves the performance of early-stage classifiers, though it may lead to a slight decrease in performance for the final classifier in some cases (e.g., LwF). However, in exemplar-based methods, we see significant accuracy improvements in intermediate layers with no degradation in any of the layers. The gains in the exemplar-based approach are likely due to the network's enhanced ability to retain knowledge during training, which aligns with our earlier findings in Section 3.1.

Our analysis shows that classifiers trained end-to-end generally achieve better accuracy in non-naive continual learning methods (a broader comparison is provided in Appendix D.3). Building on these findings, the next section further investigates how end-to-end trained ACs can be utilized to enhance both the accuracy and efficiency of the continual learning approaches.

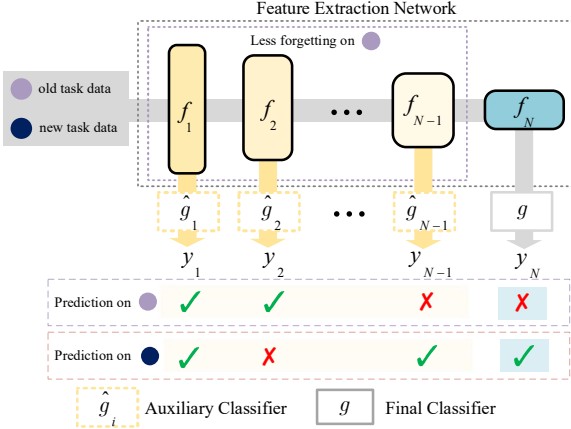

*Figure 5.* Overview of the network enhanced with ACs in continual learning. The early layers exhibit less forgetting on the old tasks, and can return the correct prediction (✓) in cases where the final classifier fails (✗), and save computations.

## 4. Enhancing CL through ACs

### 4.1. Combining from multiple classifier predictions

Our analysis in Section 3 demonstrates that auxiliary classifiers (ACs) can learn to classify different subsets of data than just a standard, single-classifier network, which hints that combining their predictions should yield improved accuracy. Therefore, we advocate the use of such multi-classifier networks in continual learning. Formally, we consider a neural network composed of backbone $f = f_N(...(f_1(x)))$ and final classifier $g$, where $f_1, ..., f_N$ are submodules in the backbone. The standard network prediction $y$ for a given input $x$ can be written as $y = g(f(x))$. We introduce additional $N - 1$ auxiliary classifiers $\hat{g}_i$ on top of the backbone sub-modules $f_1, f_2, ..., f_{N-1}$. During inference with such multi-classifier network, we obtain $N$ predictions: $y_1 = \hat{g}_1(f_1(x)), y_2 = \hat{g}_2(f_2(x)), ..., y_{N-1} = \hat{g_{N-1}}(f_{N-1}(x)), y_N = g(f(x))$ and select the prediction $y_i$ where the class predicted by the corresponding probability distribution $p_i$ has maximum confidence. Therefore, we return $y = y_{\arg\max_{i \in \{1,...,N\}} \max_k p_i^{(k)}}$, where $p_i^{(k)}$ represents the predicted probability for class $k$ in the distribution $p_i$. We refer to this simple inference paradigm as *static inference* and use it in most of the experiments, as we find it performs well across all tested settings. We show the conceptual diagram of our network in Figure 5.

### 4.2. Dynamic inference with ACs

Inspired by the early-exit models (Panda et al., 2016; Teerapittayanon et al., 2016; Kaya et al., 2019), we also consider using ACs as a means to reduce the average computational cost of the classification through *dynamic inference*. Specifically, we perform inference sequentially through the clas-

*Table 1.* The final accuracy of diverse continual learning methods enhanced with auxiliary classifiers (ACs) on CIFAR100 and ImageNet100 benchmarks. Adding ACs improves the performance of all methods across both benchmarks, demonstrating the robustness of our idea.

| Method | FT | FT+Ex | GDumb | ANCL | BiC | DER++ | ER | EWC | LwF | LODE | SSIL | Avg |
|---|---|---|---|---|---|---|---|---|---|---|---|---|
| | | | | | CIFAR100x5 | | | | | | | |
| Base | $18.68_{\pm0.31}$ | $38.35_{\pm0.86}$ | $19.09_{\pm0.44}$ | $37.71_{\pm1.14}$ | $47.66_{\pm0.43}$ | $38.96_{\pm1.38}$ | $34.55_{\pm0.21}$ | $18.95_{\pm0.29}$ | $38.26_{\pm0.98}$ | $42.82_{\pm0.84}$ | $45.62_{\pm0.16}$ | $34.60_{\pm0.20}$ |
| +AC | $\mathbf{28.18}_{\pm1.07}$ | $\mathbf{38.75}_{\pm0.26}$ | $\mathbf{23.29}_{\pm0.54}$ | $\mathbf{39.83}_{\pm1.22}$ | $\mathbf{50.40}_{\pm0.68}$ | $\mathbf{43.49}_{\pm0.73}$ | $\mathbf{39.77}_{\pm0.32}$ | $\mathbf{28.96}_{\pm1.13}$ | $\mathbf{40.55}_{\pm0.95}$ | $\mathbf{49.13}_{\pm0.35}$ | $\mathbf{48.35}_{\pm0.50}$ | $\mathbf{39.15}_{\pm0.59}$ |
| Δ | $+9.49_{\pm0.96}$ | $+0.39_{\pm0.90}$ | $+4.20_{\pm0.16}$ | $+2.12_{\pm1.03}$ | $+2.74_{\pm0.83}$ | $+4.53_{\pm2.05}$ | $+5.22_{\pm0.38}$ | $+10.02_{\pm1.39}$ | $+2.29_{\pm0.25}$ | $+6.31_{\pm0.81}$ | $+2.72_{\pm0.42}$ | $+4.55_{\pm0.38}$ |
| | | | | | CIFAR100x10 | | | | | | | |
| Base | $10.27_{\pm0.05}$ | $34.51_{\pm0.40}$ | $22.22_{\pm0.72}$ | $30.69_{\pm0.62}$ | $42.87_{\pm1.51}$ | $38.54_{\pm0.65}$ | $32.31_{\pm0.82}$ | $10.20_{\pm0.35}$ | $29.56_{\pm0.44}$ | $38.87_{\pm0.45}$ | $42.29_{\pm0.49}$ | $30.21_{\pm0.18}$ |
| +AC | $\mathbf{16.88}_{\pm1.08}$ | $\mathbf{36.97}_{\pm0.39}$ | $\mathbf{27.74}_{\pm0.73}$ | $\mathbf{31.37}_{\pm0.94}$ | $\mathbf{46.19}_{\pm1.47}$ | $\mathbf{39.64}_{\pm1.00}$ | $\mathbf{37.32}_{\pm0.28}$ | $\mathbf{19.12}_{\pm0.88}$ | $\mathbf{30.31}_{\pm1.14}$ | $\mathbf{45.67}_{\pm0.52}$ | $\mathbf{44.17}_{\pm0.28}$ | $\mathbf{34.13}_{\pm0.24}$ |
| Δ | $+6.62_{\pm1.06}$ | $+2.46_{\pm0.31}$ | $+5.52_{\pm1.13}$ | $+0.68_{\pm0.79}$ | $+3.31_{\pm2.62}$ | $+1.10_{\pm1.08}$ | $+5.01_{\pm0.98}$ | $+8.92_{\pm1.06}$ | $+0.74_{\pm0.91}$ | $+6.80_{\pm0.93}$ | $+1.88_{\pm0.77}$ | $+3.91_{\pm0.41}$ |
| | | | | | ImageNet100x5 | | | | | | | |
| Base | $23.27_{\pm0.39}$ | $44.05_{\pm0.69}$ | $21.29_{\pm0.59}$ | $60.79_{\pm0.06}$ | $62.55_{\pm0.53}$ | $45.33_{\pm2.55}$ | $38.65_{\pm0.43}$ | $23.36_{\pm0.64}$ | $59.60_{\pm0.27}$ | $49.88_{\pm0.56}$ | $60.54_{\pm0.32}$ | $44.48_{\pm0.31}$ |
| +AC | $\mathbf{34.93}_{\pm0.65}$ | $\mathbf{46.75}_{\pm0.61}$ | $\mathbf{25.30}_{\pm1.14}$ | $\mathbf{62.99}_{\pm0.30}$ | $\mathbf{65.22}_{\pm0.27}$ | $\mathbf{54.14}_{\pm0.80}$ | $\mathbf{44.46}_{\pm0.47}$ | $\mathbf{35.09}_{\pm0.17}$ | $\mathbf{61.07}_{\pm0.57}$ | $\mathbf{56.23}_{\pm0.66}$ | $\mathbf{63.89}_{\pm0.18}$ | $\mathbf{50.01}_{\pm0.13}$ |
| Δ | $+11.67_{\pm0.77}$ | $+2.71_{\pm0.85}$ | $+4.01_{\pm0.61}$ | $+2.21_{\pm0.35}$ | $+2.67_{\pm0.79}$ | $+8.81_{\pm3.34}$ | $+5.81_{\pm0.66}$ | $+11.73_{\pm0.71}$ | $+1.47_{\pm0.45}$ | $+6.35_{\pm1.22}$ | $+3.35_{\pm0.48}$ | $+5.53_{\pm0.41}$ |
| | | | | | ImageNet100x10 | | | | | | | |
| Base | $14.40_{\pm0.30}$ | $35.94_{\pm0.86}$ | $22.55_{\pm0.62}$ | $49.96_{\pm0.46}$ | $56.32_{\pm0.47}$ | $38.45_{\pm1.95}$ | $32.45_{\pm0.35}$ | $14.69_{\pm0.20}$ | $49.15_{\pm0.38}$ | $45.75_{\pm0.50}$ | $56.35_{\pm0.51}$ | $37.82_{\pm0.35}$ |
| +AC | $\mathbf{22.14}_{\pm0.16}$ | $\mathbf{39.26}_{\pm0.61}$ | $\mathbf{25.93}_{\pm0.52}$ | $\mathbf{52.07}_{\pm0.50}$ | $\mathbf{57.23}_{\pm0.87}$ | $\mathbf{45.70}_{\pm0.40}$ | $\mathbf{37.10}_{\pm1.20}$ | $\mathbf{23.25}_{\pm0.55}$ | $\mathbf{49.51}_{\pm0.71}$ | $\mathbf{51.39}_{\pm0.91}$ | $\mathbf{57.71}_{\pm0.08}$ | $\mathbf{41.93}_{\pm0.25}$ |
| Δ | $+7.74_{\pm0.37}$ | $+3.32_{\pm0.90}$ | $+3.38_{\pm0.37}$ | $+2.11_{\pm0.32}$ | $+0.91_{\pm0.42}$ | $+7.25_{\pm1.55}$ | $+4.65_{\pm0.88}$ | $+8.56_{\pm0.39}$ | $+0.36_{\pm1.05}$ | $+5.64_{\pm0.90}$ | $+1.35_{\pm0.59}$ | $+4.12_{\pm0.13}$ |

sifiers $\hat{g}_1, \hat{g}_2, ..., g$, and at each stage $i$, we compute the probability distribution $p_i$ corresponding to the prediction of $i$-th classifier. If the confidence exceeds a threshold $\lambda$, we return the corresponding prediction $y_i$. If no prediction satisfies the threshold, we use the static inference rule to determine the prediction. Formally, we define this as:

$$y = \begin{cases} y_{\min\{i \in \{1,...,N\} | \max_k p_i^{(k)} \geq \lambda\}} & \text{if such } i \text{ exists,} \\ y_{\arg\max_{i \in \{1,...,N\}} \max_k p_i^{(k)}} & \text{otherwise.} \end{cases} \quad (1)$$

By varying the confidence threshold, one can trade off the amount of computation performed by the network for slightly lower performance, which allows such a model to be deployed in settings requiring computational adaptability.

Note that our use of ACs is different from the early-exit literature, where the model accuracy usually monotonically improves when going through subsequent classifiers and the model returns the prediction of the last classifier in case no classifier can satisfy the exit threshold. As shown in Section 3, in continual learning the accuracy and quality of intermediate predictions significantly vary for different tasks, and the last classifier is not always the best one for a given subset of data. Refer to Appendix F.7 for a comparison between the performance of the standard early-exit inference rule and our method in continual learning setting.

### 4.3. AC-enhanced CL methods

To demonstrate the effectiveness of our idea, we extend several continual learning methods with auxiliary classifiers (ACs) and examine their performance. In total, we investigate ACs with the following continual learning methods: FT (Masana et al., 2022), GDUMB (Prabhu et al., 2020), EWC (Kirkpatrick et al., 2017), LwF (Li & Hoiem,

2017), ER (Chaudhry et al., 2019), DER++ (Buzzega et al., 2020), BiC (Wu et al., 2019), SSIL (Ahn et al., 2021), ANCL (Kim et al., 2023) and LODE (Liang & Li, 2024). For all the methods, we replicate the method logic (loss) across all the classifiers and do not introduce classifier-specific parameters. If the original method introduces a hyperparameter, we use the same value for this hyperparameter across all the classifiers. We also use the same batches of data for each classifier during the training. Similar to (Kaya et al., 2019), to prevent overfitting the network to the early layer classifiers we scale the total loss of each classifier according to its position so that the losses from early classifiers are weighted less than the losses for the final classifier.

## 5. Experimental results

In this section, we present the experimental results for AC-enhanced networks in CL. All our experiments are conducted with the FACIL (Masana et al., 2022) framework. We perform experiments on CIFAR100 (Krizhevsky, 2009) and ImageNet100 (the first 100 classes from ImageNet (Deng et al., 2009)), split into tasks containing different numbers of classes. Unless stated otherwise, we report average accuracy across all tasks at the end of the training. In Appendices F.1 and F.2, we additionally provide the isolated accuracies of the networks on each task across the training and the final forgetting after training on all the tasks. For all exemplar-based methods (BiC, DER++, ER, GDUMB, LODE, and SSIL), we maintain a fixed-size memory budget of 2000 exemplars, updated after each task. We report results averaged over three random seeds. More details about our experimental setup can be found in Appendix A.

## 5.1. Improved CL with ACs

First, we demonstrate the effectiveness of our approach in standard CL settings. For our main experiments, we use well-established architectures: ResNet32 for CIFAR100 and ResNet18 (He et al., 2016) for ImageNet100. In both cases, following Section 3, we use six ACs. For ResNet32, we follow the previously described AC placement, while for ResNet18, we attach ACs to the six residual blocks between the first and last one.

Additionally, in Appendices C.1 and C.2, we show that our findings extend beyond the standard 5- and 10-task splits and explore warm-start continual learning (Goswami et al., 2024), where half the data is used for the first task, as well as continual learning on more challenging, fine-grained 20- and 50-task sequences.

**Standard CL benchmarks.** We evaluate our approach on CIFAR100 and ImageNet100, each split into 5 and 10 equally sized, disjoint tasks, and present the main results in Table 1. Across all methods and settings, adding ACs consistently improves final performance, with the average relative improvement exceeding 10% of the baseline accuracy in every scenario tested. Interestingly, naive finetuning and EWC exhibit particularly strong gains, highlighting the potential of our simple yet effective idea. Exemplar-based methods (GDumb, DER++, ER, LODE) benefit more from ACs than one with additional distillation (ACNL, BiC, SSIL), which suggests that distillation may hinder ACs' ability to diversify and mitigate forgetting. Overall, our approach reliably enhances performance for all methods across all scenarios. In the next paragraph, we explore how ACs can be leveraged to also improve inference efficiency.

**Leveraging ACs for dynamic inference.** The previous section demonstrates that ACs improve performance in continual learning when performing the inference through the full network. However, ACs also can accelerate the network inference, as described in Section 4.2. To show how our approach enhances the effectiveness of continual learning methods, we evaluate selected techniques on 10- and 5-task splits of CIFAR100 and ImageNet100 using dynamic inference. We conduct the evaluation with $\lambda \in \{1, 2, ..., 100\}\%$ to assess the accuracy of the methods achieved at a given computational budget. We measure the inference cost in FLOPs, and report it relative to the non-AC network. Our results are presented in Figures 1 and 6.

AC-enhanced networks match the accuracy of the single-classifier alternatives while using only 50%-70% of the computation on CIFAR100 and 40%-60% on ImageNet100. Notably, for most methods, performance stabilizes at 70%-90% of the computational cost, which indicates that substantial computation savings are possible without sacrificing

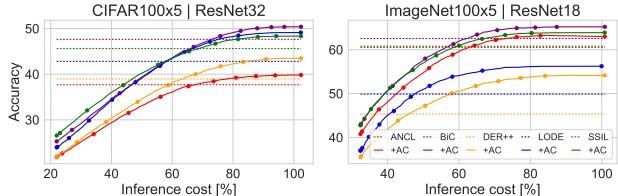

*Figure 6.* Dynamic inference on 5 tasks for AC-enhanced networks on CIFAR100 and ImageNet100. ACs improve accuracy while reducing computational cost, offering a selectable trade-off. We mark every 10% dynamic inference threshold with dots.

accuracy. To unlock those savings, it is essential to properly set the exit threshold $\lambda$. Therefore, an important factor to consider is how sensitive to this threshold is the dynamic inference. In our experiments, AC-enhanced methods consistently perform better than the baselines at the confidence thresholds above 70%, and there is no noticeable drop in the network accuracy the thresholds above 90%. These results indicate that dynamic inference can be easily tuned to provide meaningful computational savings. Moreover, we highlight how at the test-time ACs enable smooth adjustment of the average computation spent on the inference without the need for any further training. Further cost-accuracy plots for all methods from Table 1 can be found in Appendix E.1.

## 5.2. ACs with deeper networks

In this section, we conduct experimental evaluations of our approach on larger and deeper models, specifically VGG19 (Simonyan & Zisserman, 2014) and ViT (Dosovitskiy, 2020), which helps us gain a deeper understanding of how the impact of ACs relates to network size. Our results demonstrate good scaling of the ACs with network size, highlighting the robustness of our approach.

**Number of ACs with deep CNN.** We investigate the impact of the number of ACs in the deep convolutional VGG19 model, which allows us to add more classifiers. We evaluate three AC-enhanced VGG variants: 1) 18 ACs at all 18 intermediate layers, 2) 10 ACs at every other convolutional layer and both fully connected layers, and 3) 6 ACs at every fourth convolutional layer and both fully connected layers. We present the final accuracy for those setups in Table 2, with dynamic inference curves for selected methods enhanced with 18 ACs displayed in Figure 7.

In all configurations, AC-enhanced methods outperform their baseline counterparts. As expected, denser AC placements generally lead to better performance. Using more ACs appears to be a reliable strategy for improving performance, although some outliers (e.g. DER++) perform better with fewer ACs. Interestingly, for VGG19 architecture, the ACs provide a more substantial performance

*Table 2.* VGG19 enhanced with different numbers of ACs on CIFAR100. ACs integrate seamlessly with deeper network architectures, and the higher number of classifiers usually provides more significant gains. The brighter colors correspond to the better scores.

| Method | FT | FT+Ex | GDumb | ANCL | BiC | DER++ | ER | EWC | LwF | LODE | SSIL | Avg |
|---|---|---|---|---|---|---|---|---|---|---|---|---|
| | | | | | CIFAR100x5 | | | | | | | |
| Base | $18.91_{\pm0.14}$ | $42.56_{\pm0.55}$ | $26.64_{\pm1.24}$ | $40.73_{\pm0.57}$ | $52.75_{\pm0.70}$ | $45.52_{\pm0.29}$ | $33.82_{\pm0.20}$ | $18.72_{\pm0.36}$ | $39.54_{\pm0.59}$ | $46.69_{\pm0.64}$ | $47.79_{\pm0.11}$ | $37.61_{\pm0.19}$ |
| +6AC | $26.71_{\pm0.62}$ | $42.78_{\pm0.62}$ | $29.61_{\pm1.11}$ | $47.64_{\pm0.66}$ | $56.62_{\pm1.13}$ | $51.16_{\pm0.64}$ | $37.45_{\pm0.40}$ | $26.98_{\pm0.68}$ | $44.81_{\pm0.64}$ | $52.03_{\pm0.07}$ | $52.87_{\pm0.42}$ | $42.61_{\pm0.43}$ |
| +10AC | $29.16_{\pm0.23}$ | $43.05_{\pm0.45}$ | $31.36_{\pm0.73}$ | $49.12_{\pm0.70}$ | $58.05_{\pm0.44}$ | $51.03_{\pm0.23}$ | $39.06_{\pm0.70}$ | $29.40_{\pm0.32}$ | $46.51_{\pm0.52}$ | $50.39_{\pm0.65}$ | $55.30_{\pm0.32}$ | $43.86_{\pm0.22}$ |
| +18AC | $31.47_{\pm0.34}$ | $43.53_{\pm0.39}$ | $31.06_{\pm0.82}$ | $48.49_{\pm1.02}$ | $59.03_{\pm0.41}$ | $50.67_{\pm0.89}$ | $39.91_{\pm0.33}$ | $30.66_{\pm0.63}$ | $48.22_{\pm0.16}$ | $51.27_{\pm0.85}$ | $56.35_{\pm0.16}$ | $44.61_{\pm0.23}$ |
| | | | | | CIFAR100x10 | | | | | | | |
| Base | $9.52_{\pm0.17}$ | $34.20_{\pm0.48}$ | $28.79_{\pm0.66}$ | $19.50_{\pm0.97}$ | $44.30_{\pm1.70}$ | $41.88_{\pm0.44}$ | $28.85_{\pm0.83}$ | $9.37_{\pm0.43}$ | $21.04_{\pm0.79}$ | $40.08_{\pm0.52}$ | $42.49_{\pm0.73}$ | $29.09_{\pm0.18}$ |
| +6AC | $16.73_{\pm0.31}$ | $35.29_{\pm0.39}$ | $30.99_{\pm0.69}$ | $26.96_{\pm1.00}$ | $50.36_{\pm0.73}$ | $45.02_{\pm0.13}$ | $32.48_{\pm0.48}$ | $17.16_{\pm0.37}$ | $28.80_{\pm0.97}$ | $45.67_{\pm0.39}$ | $47.39_{\pm0.30}$ | $34.26_{\pm0.06}$ |
| +10AC | $18.91_{\pm0.36}$ | $36.99_{\pm0.15}$ | $31.55_{\pm0.35}$ | $29.73_{\pm0.46}$ | $52.69_{\pm0.56}$ | $43.94_{\pm0.58}$ | $33.95_{\pm0.41}$ | $19.78_{\pm0.32}$ | $31.28_{\pm0.73}$ | $46.27_{\pm0.58}$ | $48.29_{\pm1.02}$ | $35.76_{\pm0.15}$ |
| +18AC | $21.16_{\pm0.13}$ | $37.54_{\pm0.19}$ | $31.63_{\pm0.78}$ | $32.61_{\pm0.60}$ | $52.56_{\pm0.53}$ | $43.94_{\pm0.82}$ | $34.86_{\pm0.32}$ | $20.54_{\pm0.28}$ | $32.54_{\pm0.22}$ | $47.62_{\pm0.14}$ | $49.68_{\pm0.10}$ | $36.79_{\pm0.10}$ |

boost than for ResNet32, and the best 18-AC setup yields average relative improvements of approximately 20% and 25% for 5 and 10 tasks, respectively. Notably, for 10 tasks, VGG19 with ACs achieves a 2.5% accuracy improvement over AC-ResNet32 across all methods, even though the non-AC baseline VGG19 is on average 1% worse than the baseline ResNet32. This suggests the potential gains from ACs are higher in deeper networks. We also hypothesize that VGG may develop more diverse and robust classifiers than ResNet due to the absence of residual connections, which further benefits our approach.

In Figure 7, we show that the VGG19 enhanced with 18 ACs is also well-suited for dynamic inference, matching baseline performance at just 20–40% of the full model's computation for non-naive methods and achieving the maximum accuracy at as few as 40% of the computation. We present detailed results for all VGG19 methods in Appendix E.2. Additionally, in Appendices D.1 and D.2, we explore AC placement and architecture for ResNet32, and in Appendix B we quantify the training overhead of our method. We find that the impact of using more ACs is more pronounced in larger networks, where the potential for reducing inference cost is higher.

**ACs in Vision Transformers.** We further explore the compatibility of ACs with larger models by continually training base Vision Transformer (ViT[1]). To ensure a fair comparison with prior experiments, we train ViT from scratch, incorporating ACs into each transformer block (11 blocks in total). Following the standard classifier design in ViT, we perform classification using the [CLS] token and implement ACs with a single LayerNorm followed by a linear classification layer. This AC architecture is notably lightweight compared to the standard transformer block, which makes integrating ACs with transformers particularly seamless. In Figure 7, we present dynamic inference results for ViT with a selected subset of methods on the 10-task split of ImageNet100.

As highlighted in works like Pan et al. (2022); Wang et al.

---

[1]For simplicity, we refer to ViT-b-16 as ViT in this work.

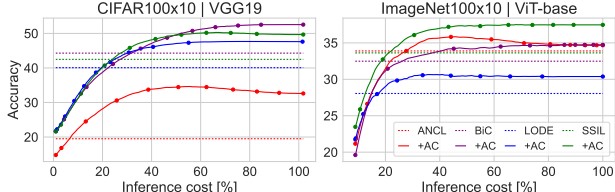

*Figure 7.* Dynamic inference plots for selected CL methods extended with ACs for VGG19 and ViT trained for 10 tasks from scratch on CIFAR100 and ImageNet100. Dynamic inference with ACs is even more beneficial with bigger networks. We show the results for all the methods in Appendices E.2 and E.3.

(2022), ViT is notoriously data-hungry. Therefore, when trained from scratch in continual learning scenarios, it tends to perform less effectively than ResNet18. However, ACs still integrate effectively with the ViT architecture, delivering superior results compared to the baseline counterparts across all methods. AC-enhanced methods achieve performance on par with their standard counterparts at particularly low 20-40% computation and quickly reach the saturation point, attaining full performance at 50% of the computational budget. These findings highlight the scalability of our approach and its compatibility with larger models, where the benefits and computational savings are even more pronounced due to richer representations.

### 5.3. ACs with wider networks

Wider networks offer richer, higher-dimensional feature representations, but they may also impose greater overhead on attached classifiers. This makes it particularly interesting to assess how network width affects the applicability of our approach. To this end, we evaluate AC-enhanced models using WideResNet16-2 (Zagoruyko & Komodakis, 2016) on CIFAR100. This architecture is shallower than ResNet-32 but twice as wide, with approximately 0.7M parameters compared to 0.46M in ResNet-32. To measure performance under our framework, we attach five additional classifiers

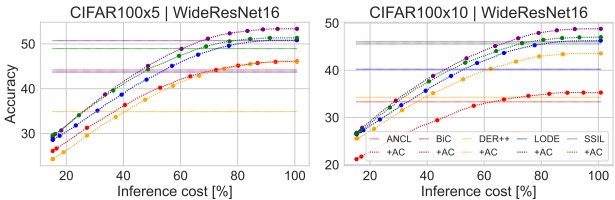

*Figure 8.* Dynamic inference results with WideResNet16.

to each ResNet block in the model. Results for the main continual learning methods are presented in Figure 8, with complementary results for other methods provided in Appendix E.4.

Consistent with the previous experiments for other architectures, AC-enhanced variants outperform their corresponding baselines across all configurations. Interestingly, in line with the recent work (Mirzadeh et al., 2022a;b) which suggests that wider networks may be less prone to forgetting, the final accuracies with WideResNet16-2 are better than with ResNet32. These findings further underscore the generality of our method, demonstrating its effectiveness even in architectures less commonly used for continual learning.

## 6. Conclusions

In this work, we show how intermediate representations in neural networks can be leveraged to improve both performance and efficiency in continual learning through auxiliary classifiers (ACs). Our analysis reveals that representations in the early layers are more stable and less prone to forgetting compared to those in the final classifier. Based on these findings, we propose attaching lightweight ACs to intermediate layers to boost network performance while enabling faster inference through dynamic layer skipping. Our experiments across diverse continual learning methods show an average 10% relative performance improvement over standard single-classifier models on CIFAR100 and ImageNet100, consistent with different architectures such as ResNets, VGG19, and ViT. Additionally, through dynamic inference, AC-enhanced models reduce the computational cost of the inference by up to 60% without sacrificing accuracy. Overall, our approach provides a promising solution to address both performance and efficiency challenges in resource-constrained continual learning environments.

**Discussion.** Our findings demonstrate that adding ACs improves overall continual learning performance. The gains from ACs might seem counterintuitive at first, especially since early ACs are less accurate and also not immune to forgetting. However, different ACs learn to classify samples based on different features. In particular, early classifiers rely on shared, general features, which makes them more

robust across tasks and capable of correctly classifying distinct subsets of data, even if their overall accuracy is lower. As a result, using multiple ACs introduces both diversity and redundancy in predictions - if one classifier forgets, another may still succeed. This redundancy, combined with confidence-based early exits, makes ACs a viable tool for continual learning and helps explain how they can improve not only efficiency but also performance.

**Reproducibility.** The code used to run experiments in this paper is publicly available at `https://github.com/fszatkowski/cl-auxiliary-classifiers`.

**Limitations.** ACs add slight training and memory overhead, which varies by method and architecture. This study focused on computer vision and classification, leaving opportunities for exploration in other modalities and tasks.

## Impact Statement

This work aims to advance the scientific understanding of machine learning, particularly in the field of continual learning. Our primary contribution is to enhance the fundamental principles of AI research, fostering progress within the academic and scientific community. While our work has the potential to influence broader applications, we emphasize that its intended purpose is purely scientific. We do not take responsibility for any unintended consequences that may arise from its application beyond this scope, especially in an era of rapid AI and deep learning advancements.

## Acknowledgements

This work is supported by National Centre of Science (NCP, Poland) Grants No. 2022/45/B/ST6/02817, 2024/53/N/ST6/03078, and 2023/51/D/ST6/02846. Yaoyue Zheng acknowledges the China Scholarship Council (CSC) No.202406280387. This work was supported by Horizon Europe Programme under GA no. 101120237, project "ELIAS: European Lighthouse of AI for Sustainability". We gratefully acknowledge Polish high-performance computing infrastructure PLGrid (HPC Center: ACK Cyfronet AGH) for providing computer facilities and support within computational grant no. PLG/2024/017385. We acknowledge the Spanish project PID2022-143257NB-I00, financed by MCIN/AEI/10.13039/501100011033 and FEDER, and Funded by the European Union ELLIOT project. Bartłomiej Twardowski acknowledges the grant RYC2021-032765-I.

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

# Appendix

## A. AC architecture and training details

In our main experiments, we follow the insights from (Kaya et al., 2019) and (Wójcik et al., 2023) in our design of multi-classifier networks. We place the ACs after layers that perform roughly 15%, 30%, 45%, 60%, 75%, 90% of the computations of the full network (L1.B3, L1.B5, L2.B2, L2.B4, L3.B1, L3.B3, L3.B5 for ResNet32, and blocks B2-B7 for ResNet18). For convolutional networks, ACs are composed of pooling layers to reduce the input size for the fully connected networks that produce the predictions. For experiments on ViT, we apply a fully connected classifier on top of the LayerNorm layer on the first token. All the classifiers in our model are composed of heads for each task, and we add a new head upon encountering a new task.

Our main objective used for training the network on any given task is a weighted sum of losses for each classifier. For continual learning methods, we use the additional losses alongside the cross-entropy and weigh the total loss. We train the model for each task jointly with all the ACs, updating all the parameters of the network. We follow the weight scheduler from (Kaya et al., 2019) and progressively increase loss weights for different ACs over the training phase to the values matching their computational cost (e.g. the weight for the first classifier for ResNet32 would increase up to 0.15, for the second classifier to 0.30, and so on). For ResNet18 we use 6 ACs and set weights to $[0.3, 0.4, 0.55, 0.65, 0.8, 0.9]$, as the network contains only 8 blocks whose computational cost distributes approximately like that. For experiments with 12 ACs, we attach classifiers to all blocks L1.B3-L3.B4 and interpolate the weights from the standard setting. For 3 ACs, we use blocks L1.B3, L2.B2, and L3.B1 with weights $[0.15, 0.45, 0.75]$. When training ViT or VGG networks, for each model block we use multiplies of a given base weight (e.g. 0.08 for ViT-base, 0.05 or 0.09 for 18 and 10 AC setup for VGG19). For example, we set the AC weights for 11 ACs in ViT as $[0.08, 0.16, ..., 0.80, 0.88]$. Different loss weights for each AC serve to stabilize the training and mitigate overfitting in the earlier layers, which may have lower learning capacity.

We train the ResNet32 models on CIFAR100 for 200 epochs on each task, using SGD optimizer with a batch size of 128 with a learning rate initialized to 0.1 and decayed by a rate of 0.1 at the 60th, 120th, and 160th epochs. For training ResNet18 on ImageNet100, we change the scheduler to cosine with a linear warmup and train for 100 epochs with 5 epochs of warmup, as we find it to converge to similar results in a shorter time. For ViT, we use AdamW and train each task for 100 epochs with a learning rate of 0.01 and batch size of 64. We also use a cosine scheduler with a linear warmup for 5 epochs. We use a fixed memory of 2000 exemplars selected with herding (Rebuffi et al., 2017). For ER each batch is balanced between old and new data, and for SSIL we use a 4:1 ratio of new to old data. Otherwise, for other exemplar-based methods, we follow the standard FACIL procedure for exemplars and just add them to the training data without any balancing.

## B. Training time and memory overhead disucssion

Introducing Auxiliary Classifiers (ACs) into the network introduces additional computational and memory overhead during training, due to the extra gradient computations and memory required by the classifiers. In this section, we quantify this overhead by evaluating the impact of ACs on training ResNet-32 on CIFAR-100 with various CL methods. We train the model with different numbers of ACs, where "0 ACs" corresponds to the baseline model without any auxiliary classifiers. For each configuration, we report average training time (in hours) and peak GPU memory usage (in GB) across three random seeds. Training time is measured separately for 5-task and 10-task incremental setups, while peak memory usage is independent of the number of tasks. We present the results in Tables 3 and 4.

In the standard setup used in the main paper (6 ACs), we observe a roughly 50% increase in training time and a 10% increase in peak memory usage compared to the baseline. However, it is important to note that these results represent the worst-case overhead across all our experiments, since ResNet-32 is the smallest model we evaluate, and the relative impact of the added ACs on parameter count and computational load is the most pronounced for this model. As shown in Figures 1 and 7, the total inference cost with ACs exceeds the baseline by approximately 3% for ResNet-32, whereas for larger models like ViT, this overhead reduces to around 1%. While these overheads are non-negligible, they are not prohibitive in practical settings. Class-incremental learning is typically performed in an offline setting without real-time constraints, making the additional training costs acceptable in exchange for the performance gains achieved by ACs. Moreover, we did not specifically optimize for training efficiency, and these overheads could be further reduced if necessary in the downstream applications.

*Table 3.* Training time (in hours) for CIFAR-100 under 5-task and 10-task setups, reported as (5-task / 10-task).

| ACs | FT | FT+Ex | GDumb | ANCL | BiC | ER | EWC | LwF | LODE | SSIL | Avg |
|---|---|---|---|---|---|---|---|---|---|---|---|
| 0 | 1.1 / 1.4 | 1.3 / 1.7 | 0.6 / 1.2 | 2.1 / 2.8 | 1.4 / 2.0 | 2.4 / 2.5 | 1.2 / 1.4 | 1.2 / 1.4 | 3.0 / 3.3 | 1.7 / 1.8 | 1.6 / 1.9 |
| 3 | 1.3 / 1.6 | 1.5 / 2.0 | 0.8 / 1.5 | 2.7 / 3.6 | 1.8 / 2.5 | 2.8 / 3.2 | 1.5 / 1.9 | 1.5 / 1.9 | 3.6 / 4.2 | 2.1 / 2.3 | 2.0 / 2.5 |
| 6 | 1.6 / 1.9 | 1.7 / 2.4 | 1.1 / 1.7 | 3.3 / 4.4 | 2.1 / 3.0 | 3.3 / 3.8 | 1.8 / 2.2 | 1.8 / 2.4 | 4.3 / 5.2 | 2.5 / 2.9 | 2.4 / 3.0 |
| 12 | 2.0 / 2.4 | 2.3 / 3.3 | 1.5 / 2.3 | 4.5 / 6.8 | 2.9 / 4.2 | 4.0 / 4.8 | 2.5 / 3.0 | 2.5 / 3.5 | 5.5 / 6.9 | 3.2 / 3.9 | 3.1 / 4.1 |

*Table 4.* Peak GPU memory usage (in GB) across methods and AC configurations.

| ACs | FT | FT+Ex | GDumb | ANCL | BiC | ER | EWC | LwF | LODE | SSIL | Avg |
|---|---|---|---|---|---|---|---|---|---|---|---|
| 0 | 2.10 | 2.10 | 2.39 | 2.10 | 2.09 | 2.10 | 2.10 | 2.10 | 2.10 | 2.18 | 2.14 |
| 3 | 2.25 | 2.18 | 2.56 | 2.29 | 2.22 | 2.18 | 2.56 | 2.22 | 2.22 | 2.29 | 2.30 |
| 6 | 2.41 | 2.27 | 2.73 | 2.47 | 2.32 | 2.27 | 2.73 | 2.30 | 2.31 | 2.41 | 2.42 |
| 12 | 2.72 | 2.43 | 3.03 | 2.82 | 2.50 | 2.43 | 3.02 | 2.50 | 2.49 | 2.65 | 2.66 |

# C. ACs in more CL settings

In this section, we prove the robustness of our idea on additional benchmarks in warm-start continual learning and two setups with more tasks (20 and 50) on CIFAR100.

## C.1. Warm-start continual learning

Table 5. Adding auxiliary classifiers (ACs) is beneficial to the final network accuracy when training on CIFAR100 warm-start scenario with 50 classes in the first task.

| Method | FT | FT+Ex | GDumb | ANCL | BiC | DER++ | ER | EWC | LwF | LODE | SSIL | Avg |
|---|---|---|---|---|---|---|---|---|---|---|---|---|
| | | | | | | CIFAR100x6 | | | | | | |
| Base | $16.18_{\pm0.65}$ | $\mathbf{40.38}_{\pm0.75}$ | $17.38_{\pm0.33}$ | $42.85_{\pm1.07}$ | $46.60_{\pm1.15}$ | $38.49_{\pm0.14}$ | $\mathbf{38.11}_{\pm0.10}$ | $17.08_{\pm1.11}$ | $42.72_{\pm0.60}$ | $42.28_{\pm0.46}$ | $46.78_{\pm0.15}$ | $35.35_{\pm0.18}$ |
| +AC | $\mathbf{22.37}_{\pm1.28}$ | $38.12_{\pm0.77}$ | $\mathbf{22.60}_{\pm0.31}$ | $\mathbf{43.97}_{\pm0.41}$ | $\mathbf{48.75}_{\pm0.17}$ | $\mathbf{43.55}_{\pm0.54}$ | $37.81_{\pm0.58}$ | $\mathbf{25.49}_{\pm0.97}$ | $\mathbf{43.45}_{\pm0.74}$ | $\mathbf{44.95}_{\pm0.28}$ | $\mathbf{48.97}_{\pm0.28}$ | $\mathbf{38.18}_{\pm0.28}$ |
| Δ | $+6.19_{\pm1.72}$ | $-2.26_{\pm0.35}$ | $+5.22_{\pm0.19}$ | $+1.12_{\pm1.35}$ | $+2.15_{\pm1.23}$ | $+5.06_{\pm0.66}$ | $-0.30_{\pm0.68}$ | $+8.40_{\pm0.71}$ | $+0.72_{\pm1.13}$ | $+2.67_{\pm0.28}$ | $+2.19_{\pm0.36}$ | $+2.83_{\pm0.11}$ |
| | | | | | | CIFAR100x11 | | | | | | |
| Base | $7.90_{\pm0.30}$ | $36.41_{\pm1.06}$ | $16.55_{\pm0.41}$ | $33.86_{\pm0.11}$ | $42.38_{\pm0.64}$ | $35.24_{\pm0.90}$ | $34.86_{\pm0.56}$ | $8.01_{\pm0.88}$ | $32.13_{\pm0.72}$ | $38.17_{\pm0.17}$ | $41.46_{\pm0.84}$ | $29.73_{\pm0.04}$ |
| +AC | $\mathbf{11.91}_{\pm1.59}$ | $\mathbf{36.80}_{\pm0.45}$ | $\mathbf{22.73}_{\pm0.74}$ | $\mathbf{34.94}_{\pm0.95}$ | $\mathbf{45.37}_{\pm0.44}$ | $\mathbf{40.46}_{\pm0.55}$ | $\mathbf{36.80}_{\pm0.53}$ | $\mathbf{16.05}_{\pm0.96}$ | $\mathbf{35.31}_{\pm1.42}$ | $\mathbf{40.97}_{\pm0.22}$ | $\mathbf{45.70}_{\pm0.59}$ | $\mathbf{33.37}_{\pm0.31}$ |
| Δ | $+4.00_{\pm1.48}$ | $+0.39_{\pm0.63}$ | $+6.17_{\pm0.38}$ | $+1.08_{\pm0.85}$ | $+2.99_{\pm0.21}$ | $+5.22_{\pm1.09}$ | $+1.94_{\pm1.07}$ | $+8.04_{\pm0.67}$ | $+3.18_{\pm0.77}$ | $+2.80_{\pm0.35}$ | $+4.24_{\pm1.10}$ | $+3.64_{\pm0.35}$ |

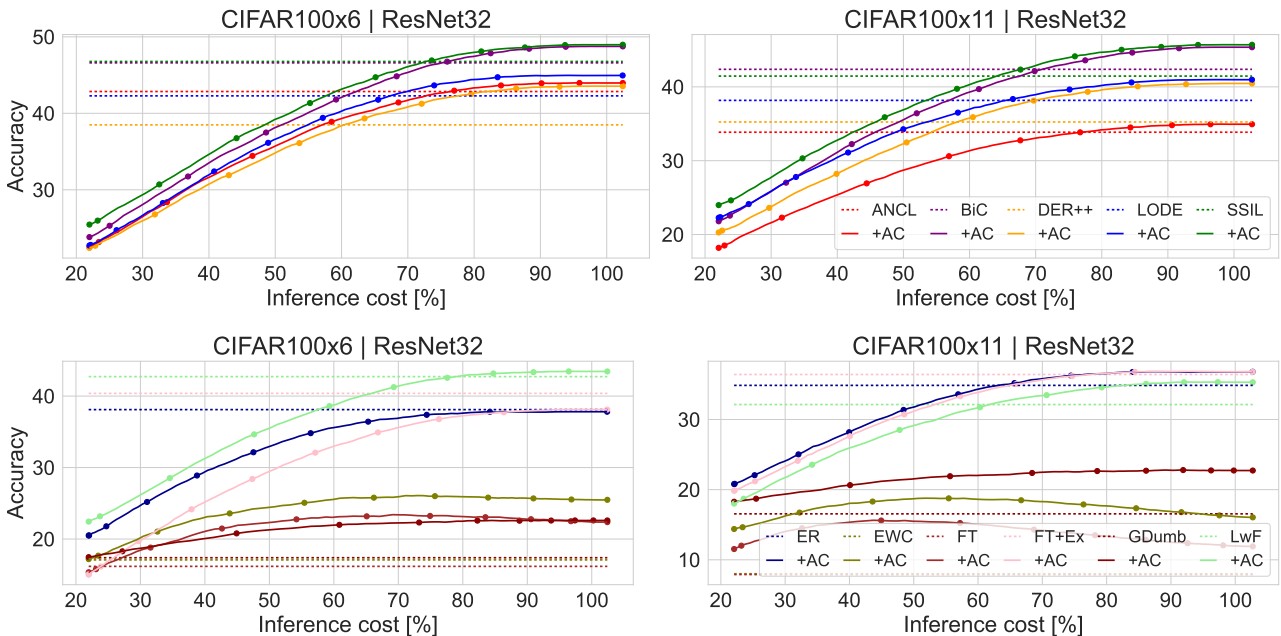

Figure 9. Dynamic inference results for CL methods enhanced with ACs in warm start scenario on CIFAR100.

A common scenario in continual learning is warm-start (Magistri et al., 2024; Goswami et al., 2024), which simulates starting from a pre-trained model. This scenario is an interesting study for continual learning due to the practical benefits of pre-trained models and the differing learning dynamics. To evaluate our model in a warm-start scenario, we train on CIFAR100, using 50 classes for the first task to simulate a pre-training phase. The remaining classes are split evenly into 5 or 10 tasks, referred to as the 6-task and 11-task splits. We conduct experiments with a new task split and results are shown in Table 5. We observe a general improvement in the warm-start setting, except for ER and FT+Ex, which show different performances on the 6-task and 11-task splits. In Figure 9, we also show the effectiveness of dynamic inference under the warm-start scenario.

*Table 6.* Additional results for AC-enhanced methods with longer sequences of tasks on CIFAR100.

| Method | FT | FT+Ex | GDumb | ANCL | BiC | DER++ | ER | EWC | LwF | LODE | SSIL | Avg |
|---|---|---|---|---|---|---|---|---|---|---|---|---|
| | | | | | CIFAR100x20 | | | | | | | |
| Base | $4.72_{\pm0.75}$ | $32.35_{\pm0.26}$ | $23.68_{\pm1.08}$ | $19.34_{\pm0.32}$ | $38.81_{\pm1.02}$ | $34.50_{\pm0.33}$ | $30.94_{\pm0.53}$ | $5.51_{\pm0.36}$ | $18.97_{\pm1.20}$ | $37.90_{\pm0.37}$ | $36.86_{\pm1.21}$ | $25.78_{\pm0.32}$ |
| +AC | $\mathbf{7.33}_{\pm0.45}$ | $\mathbf{37.16}_{\pm0.63}$ | $\mathbf{30.11}_{\pm0.36}$ | $\mathbf{20.87}_{\pm0.78}$ | $\mathbf{42.03}_{\pm0.18}$ | $\mathbf{36.45}_{\pm0.81}$ | $\mathbf{36.10}_{\pm0.07}$ | $\mathbf{9.70}_{\pm0.18}$ | $\mathbf{19.63}_{\pm0.75}$ | $\mathbf{41.85}_{\pm0.49}$ | $\mathbf{39.78}_{\pm0.24}$ | $\mathbf{29.18}_{\pm0.10}$ |
| Δ | $+2.61_{\pm0.61}$ | $+4.81_{\pm0.86}$ | $+6.43_{\pm0.98}$ | $+1.53_{\pm0.91}$ | $+3.22_{\pm1.01}$ | $+1.95_{\pm1.05}$ | $+5.16_{\pm0.60}$ | $+4.19_{\pm0.40}$ | $+0.66_{\pm1.69}$ | $+3.95_{\pm0.65}$ | $+2.92_{\pm1.03}$ | $+3.40_{\pm0.22}$ |
| | | | | | CIFAR100x50 | | | | | | | |
| Base | $0.97_{\pm0.51}$ | $21.60_{\pm0.88}$ | $13.48_{\pm0.89}$ | $\mathbf{5.88}_{\pm0.21}$ | $24.01_{\pm0.42}$ | $16.79_{\pm1.38}$ | $18.35_{\pm0.38}$ | $1.63_{\pm0.55}$ | $5.09_{\pm0.51}$ | $23.56_{\pm1.82}$ | $22.66_{\pm0.39}$ | $14.00_{\pm0.22}$ |
| +AC | $\mathbf{1.64}_{\pm0.19}$ | $\mathbf{26.39}_{\pm0.16}$ | $\mathbf{17.63}_{\pm0.90}$ | $5.54_{\pm0.25}$ | $\mathbf{29.39}_{\pm0.73}$ | $\mathbf{22.59}_{\pm0.39}$ | $\mathbf{22.96}_{\pm0.29}$ | $\mathbf{2.12}_{\pm0.02}$ | $\mathbf{5.40}_{\pm0.60}$ | $\mathbf{28.53}_{\pm1.29}$ | $\mathbf{25.77}_{\pm0.74}$ | $\mathbf{17.09}_{\pm0.27}$ |
| Δ | $+0.67_{\pm0.67}$ | $+4.79_{\pm0.88}$ | $+4.15_{\pm0.60}$ | $-0.34_{\pm0.19}$ | $+5.38_{\pm0.41}$ | $+5.80_{\pm1.01}$ | $+4.61_{\pm0.52}$ | $+0.49_{\pm0.57}$ | $+0.31_{\pm0.88}$ | $+4.97_{\pm2.06}$ | $+3.11_{\pm1.12}$ | $+3.08_{\pm0.47}$ |

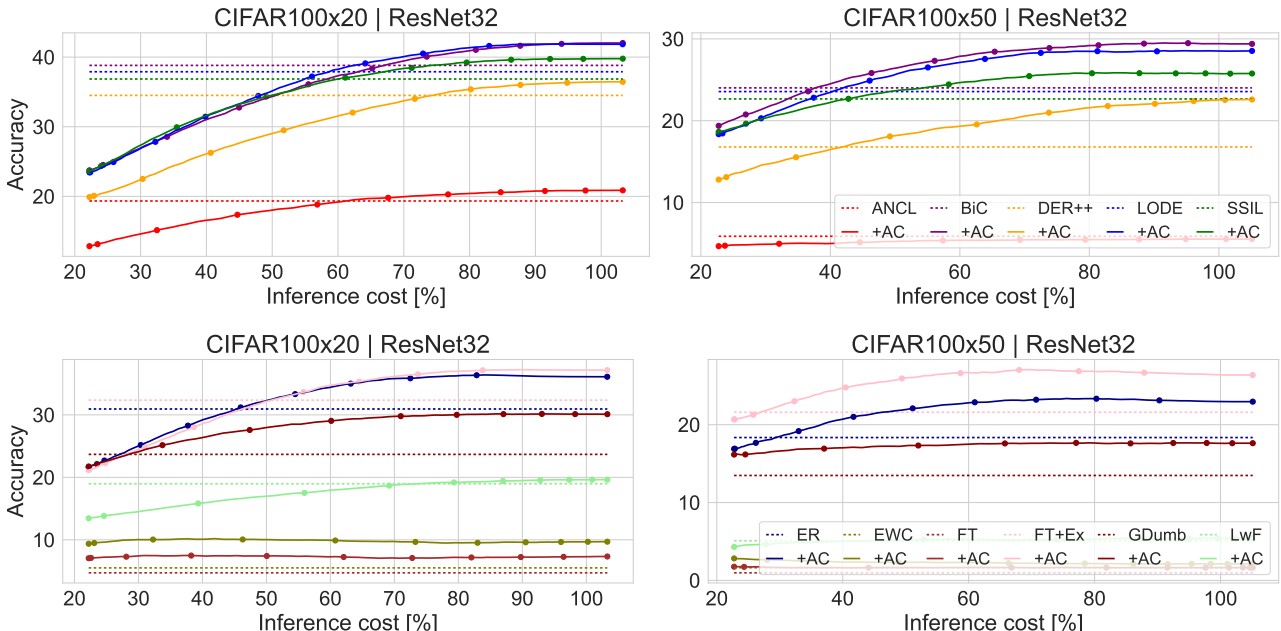

*Figure 10.* Dynamic inference results for CL methods enhanced with ACs on CIFAR100 split into long, 20 and 50 tasks sequences.

### C.2. AC-enhanced methods on longer task sequences

In Table 6 we present results for experiments on 20 and 50 equally split tasks on CIFAR100, following the setup from Section 5.1. For the 50-task split, we use a growing memory of 20 exemplars instead of a constant memory of 2000 due to the early tasks containing less samples than the memory limit. Non-replay-based methods perform poorly on longer sequences of tasks, especially on 50 tasks, but ACs robustly enhance the method performance in all tested scenarios. In Figure 10, we also demonstrate the performance of AC-enhanced methods with dynamic inference.

# D. Ablation studies on AC design

In this section, we perform ablation studies on AC design in the setup from Section 5.1 for CIFAR100. In Appendices D.1 and D.2 we explore the impact of the number of the ACs and alternative AC architectures from Wójcik et al. (2023). In Appendix D.3 we compare models trained end-to-end and obtained through linear probing, and in Appendix D.4 we show the performance of networks with only a single classifier to provide the intuition behind the classifier behaviour.

## D.1. Number of ACs

We vary the number of classifiers for the 6 AC setup from the main paper: we either drop half of them or insert an extra one between the existing classifiers. We measure the improvement obtained upon the baseline and the additional computational cost incurred by 3, 6, and 12 ACs and show the results in Table 7 and Figure 11. While the optimal setup varies across different continual learning methods, the addition of ACs is universally beneficial to performance.

*Table 7.* Difference w.r.t. baseline single-classifier methods when using a different number of auxiliary classifiers (ACs). ACs robustly improve the final accuracy of continual learning methods, regardless of the number of classifiers used.

| | FLOPS | FT | FT+Ex | GDumb | ANCL | BiC | ER | EWC | LwF | LODE | SSIL | Avg |
|---|---|---|---|---|---|---|---|---|---|---|---|---|
| NoAC | 69.90M (1x) | | | | | | | | | | | |
| | | | | | | CIFAR100x5 | | | | | | |
| 3AC | 70.72M (1.01x) | +7.61$_{\pm0.68}$ | +0.70$_{\pm0.83}$ | +5.08$_{\pm0.87}$ | +2.14$_{\pm1.05}$ | +2.62$_{\pm0.60}$ | +4.63$_{\pm0.38}$ | +8.94$_{\pm0.40}$ | +0.95$_{\pm1.18}$ | +4.19$_{\pm0.63}$ | +2.65$_{\pm0.58}$ | +3.95$_{\pm0.39}$ |
| 6AC | 71.55M (1.02x) | **+9.49**$_{\pm0.96}$ | +0.39$_{\pm0.90}$ | +4.20$_{\pm0.16}$ | +2.12$_{\pm1.03}$ | +2.74$_{\pm0.83}$ | **+5.22**$_{\pm0.38}$ | **+10.02**$_{\pm1.39}$ | +2.29$_{\pm0.25}$ | **+6.31**$_{\pm0.81}$ | +2.72$_{\pm0.42}$ | +4.55$_{\pm0.43}$ |
| 12AC | 72.97M (1.04x) | +9.09$_{\pm0.81}$ | **+1.67**$_{\pm1.28}$ | **+5.15**$_{\pm0.08}$ | **+2.45**$_{\pm0.74}$ | **+3.57**$_{\pm0.47}$ | +4.46$_{\pm0.54}$ | +9.97$_{\pm0.35}$ | **+2.75**$_{\pm1.26}$ | +5.43$_{\pm0.65}$ | **+2.92**$_{\pm0.53}$ | **+4.74**$_{\pm0.20}$ |
| | | | | | | CIFAR100x10 | | | | | | |
| 3AC | 70.84M (1.01x) | +6.48$_{\pm0.43}$ | **+3.05**$_{\pm0.99}$ | +5.74$_{\pm0.47}$ | +1.71$_{\pm0.31}$ | +2.85$_{\pm2.19}$ | +4.44$_{\pm1.00}$ | +7.92$_{\pm0.61}$ | +0.53$_{\pm0.44}$ | +6.13$_{\pm0.23}$ | +2.02$_{\pm1.69}$ | +4.09$_{\pm0.42}$ |
| 6AC | 71.77M (1.03x) | **+6.62**$_{\pm1.06}$ | +2.46$_{\pm0.31}$ | +5.52$_{\pm1.13}$ | +0.68$_{\pm0.79}$ | +3.31$_{\pm2.62}$ | **+5.01**$_{\pm0.98}$ | **+8.92**$_{\pm1.06}$ | +0.74$_{\pm0.91}$ | **+6.80**$_{\pm0.93}$ | +1.88$_{\pm0.77}$ | **+4.20**$_{\pm0.47}$ |
| 12AC | 73.36M (1.05x) | +4.63$_{\pm1.46}$ | +2.59$_{\pm1.19}$ | **+6.12**$_{\pm0.78}$ | **+1.85**$_{\pm1.49}$ | **+3.98**$_{\pm2.01}$ | +4.95$_{\pm1.05}$ | +6.57$_{\pm0.97}$ | **+1.14**$_{\pm0.38}$ | +6.68$_{\pm0.79}$ | **+1.98**$_{\pm0.91}$ | +4.05$_{\pm0.60}$ |

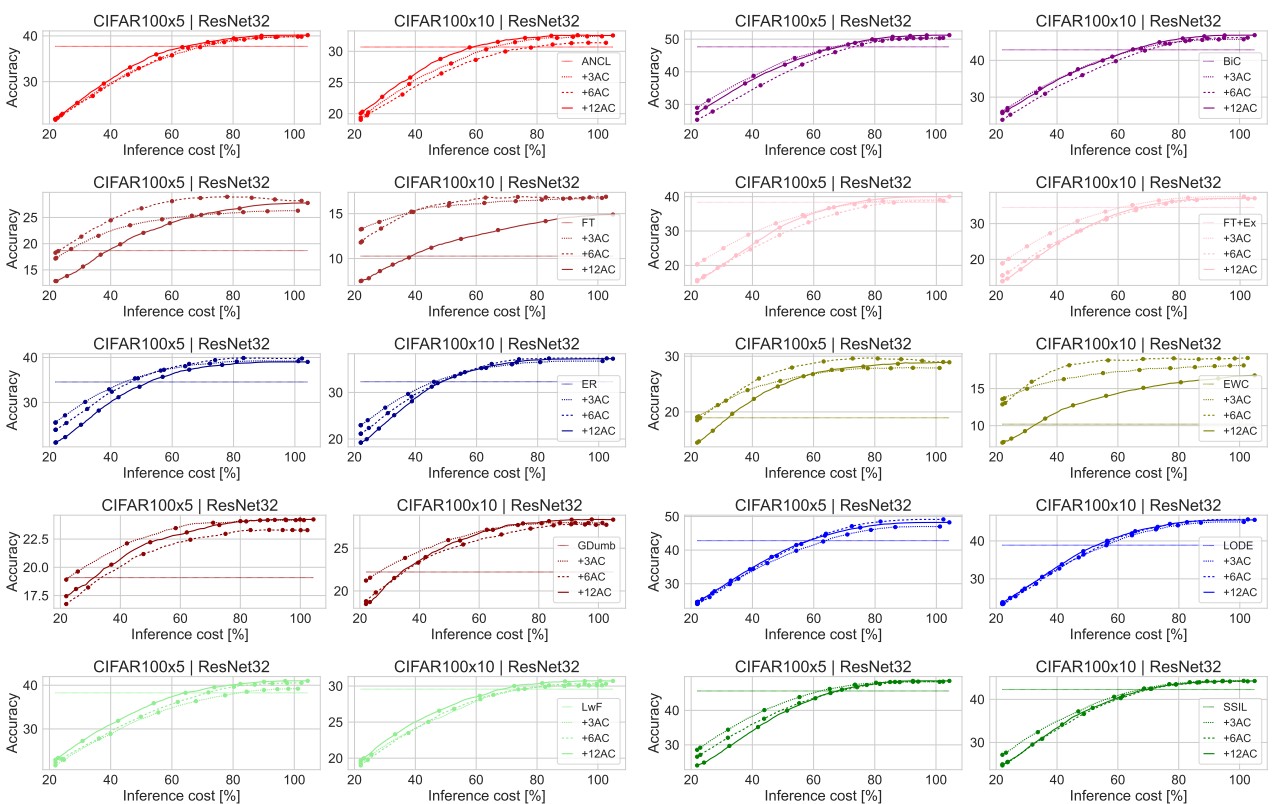

*Figure 11.* Dynamic inference experiments for ResNet32 on CIFAR100 enhanced with different number of ACs.

## D.2. AC architecture

In our main work, we investigate a simple setup with independent classifiers. Early-exit works such as (Wójcik et al., 2023) propose more complex dynamic architectures, where subsequent classifiers are connected and their predictions are combined through a weighted ensemble. Those architectures induce only a slight parameter and computation overhead, but in a standard supervised learning setting can improve the performance of intermediate classifiers through sharing the knowledge between them. We investigate those architectures in continual learning on the set of methods analyzed in previous sections on split CIFAR100 benchmarks and present the results in Table 8, alongside dynamic inference plots in Figure 12. Similar to the AC density ablation, we do not observe a clear improvement from changing the setup. We hypothesize that connecting the classifiers makes them no longer independent, which negates the benefits yielded in continual learning by the classifier diversity.

*Table 8.* Difference w.r.t. baseline single-classifier methods when using a different auxiliary classifier architecture: cascading (C) and ensebling (E) from Wójcik et al. (2023). Similar to Table 7, ACs universally improve the accuracy with small differences in performance between the architectures.

| Method | FT | FT+Ex | GDumb | ANCL | BiC | ER | EWC | LwF | LODE | SSIL | Avg |
|---|---|---|---|---|---|---|---|---|---|---|---|
| | | | | | CIFAR100x5 | | | | | | |
| AC | $+5.70_{\pm5.24}$ | $+0.24_{\pm0.67}$ | $+2.52_{\pm2.30}$ | $+1.27_{\pm1.37}$ | $+1.65_{\pm1.61}$ | $+3.13_{\pm2.87}$ | $+6.01_{\pm5.57}$ | $+1.37_{\pm1.27}$ | $+3.79_{\pm3.50}$ | $+1.63_{\pm1.52}$ | $+2.73_{\pm2.51}$ |
| AC+C | $+5.41_{\pm4.99}$ | $-0.72_{\pm0.70}$ | $+2.49_{\pm2.27}$ | $+1.20_{\pm1.37}$ | $+1.16_{\pm1.06}$ | $+2.57_{\pm2.37}$ | $+6.16_{\pm5.62}$ | $+0.66_{\pm0.77}$ | $+2.89_{\pm2.66}$ | $+1.26_{\pm1.23}$ | $+2.31_{\pm2.12}$ |
| AC+E | $+6.47_{\pm5.91}$ | $+0.38_{\pm1.03}$ | $+2.00_{\pm1.83}$ | $+1.28_{\pm39.62}$ | $+1.30_{\pm1.20}$ | $+2.32_{\pm2.16}$ | $+6.58_{\pm6.01}$ | $+1.14_{\pm1.18}$ | $+2.79_{\pm2.61}$ | $+1.29_{\pm1.23}$ | $+2.56_{\pm2.11}$ |
| | | | | | CIFAR100x10 | | | | | | |
| AC | $+6.62_{\pm1.06}$ | $+2.46_{\pm0.31}$ | $+5.52_{\pm1.13}$ | $+0.68_{\pm0.79}$ | $+3.31_{\pm2.62}$ | $+5.01_{\pm0.98}$ | $+8.92_{\pm1.06}$ | $+0.74_{\pm0.91}$ | $+6.80_{\pm0.93}$ | $+1.88_{\pm0.77}$ | $+4.20_{\pm0.47}$ |
| AC+C | $+6.98_{\pm1.05}$ | $+2.63_{\pm0.92}$ | $+5.56_{\pm0.96}$ | $+1.60_{\pm0.70}$ | $+3.63_{\pm1.51}$ | $+5.19_{\pm1.08}$ | $+8.35_{\pm0.39}$ | $+1.27_{\pm0.47}$ | $+5.45_{\pm0.17}$ | $+2.53_{\pm0.56}$ | $+4.32_{\pm0.16}$ |
| AC+E | $+7.35_{\pm0.14}$ | $+3.27_{\pm0.74}$ | $+5.09_{\pm0.40}$ | $+2.28_{\pm0.21}$ | $+3.68_{\pm2.16}$ | $+4.77_{\pm0.52}$ | $+8.62_{\pm1.12}$ | $+1.18_{\pm0.30}$ | $+5.85_{\pm0.68}$ | $+1.53_{\pm0.67}$ | $+4.36_{\pm0.33}$ |

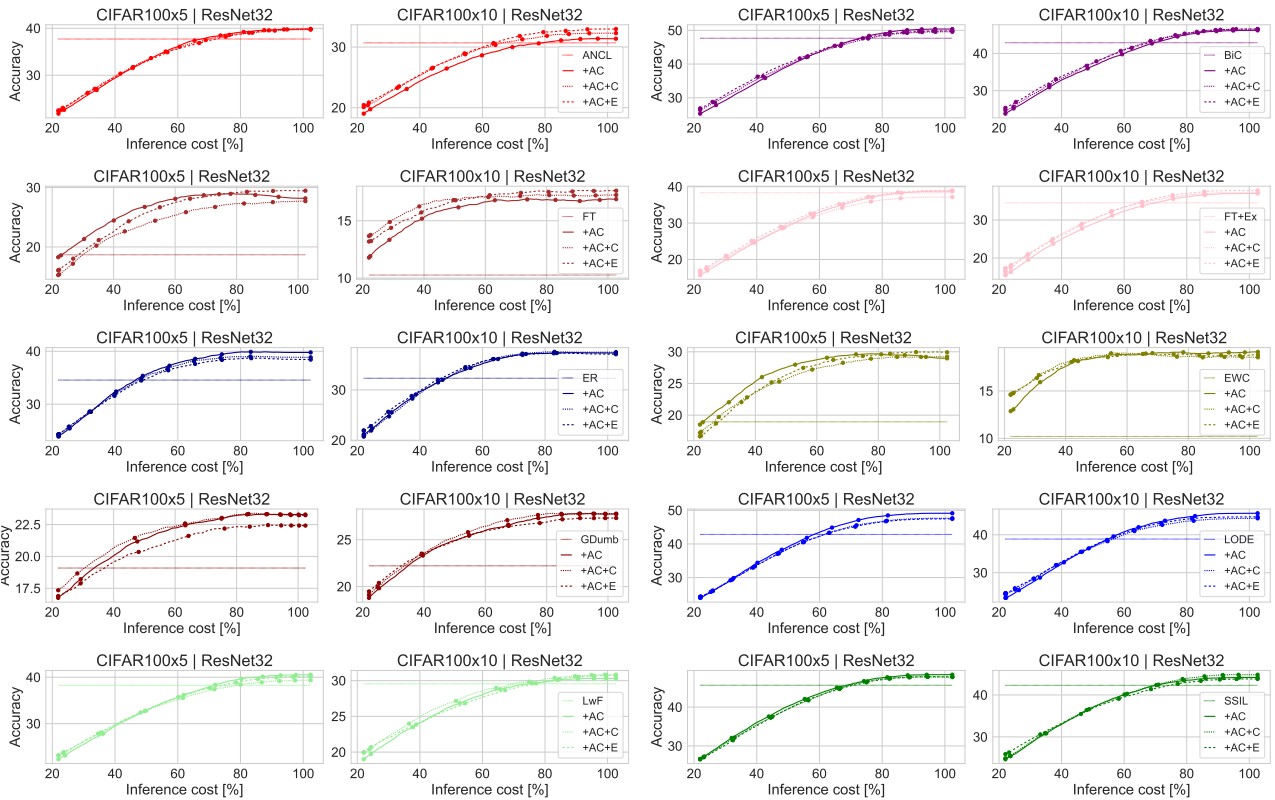

*Figure 12.* Dynamic inference experiments for ResNet32 on CIFAR100 enhanced with different types of classifiers.

## D.3. Comparison between end-to-end training and linear probing

In Section 3.4, we advocate for training the network and ACs jointly with enabled gradient propagation, as it leads to better performance of individual classifiers. In this section, we investigate the final performance of linear probing classifiers in comparison with jointly trained ACs on CIFAR100. For ACs use the same setup as in Section 5.1, and in the case of linear probing the only difference is that the classifiers are trained without gradient propagation. We show the final performance of both settings in Table 9, and also demonstrate their cost-accuracy characteristics in Figure 13. Aside from distillation-based exemplar-free methods, ACs outperform probing accuracy but probing also outperforms baselines most of the time. Dynamic accuracy curves also highlight that end-to-end training generally allows AC methods to achieve greater accuracy at lower computational costs due to the ability to learn better early classifiers.

*Table 9.* Comparison between final results when using intermediate classifiers trained together with the network (AC) or trained with linear probing (LP). Training classifiers together generally yields better performance, with the only noticeable exception being exemplar-free distillation-based methods (ANCL and LwF), which could be caused by significant variance in per-task accuracy of intermediate classifiers.

| Method | FT | FT+Ex | GDumb | ANCL | BiC | ER | EWC | LwF | LODE | SSIL | Avg |
|---|---|---|---|---|---|---|---|---|---|---|---|
| | | | | | CIFAR100x5 | | | | | | |
| Base | $18.68_{\pm0.31}$ | $38.35_{\pm0.86}$ | $19.09_{\pm0.44}$ | $37.71_{\pm1.14}$ | $47.66_{\pm0.43}$ | $34.55_{\pm0.21}$ | $18.95_{\pm0.29}$ | $38.26_{\pm0.98}$ | $42.82_{\pm0.84}$ | $45.62_{\pm0.16}$ | $34.17_{\pm0.27}$ |
| +LP | $26.82_{\pm1.19}$ | $36.83_{\pm1.27}$ | $22.56_{\pm0.63}$ | $\mathbf{43.60_{\pm0.19}}$ | $49.62_{\pm0.07}$ | $38.47_{\pm0.78}$ | $28.13_{\pm1.11}$ | $\mathbf{41.13_{\pm0.33}}$ | $46.35_{\pm0.45}$ | $47.33_{\pm0.60}$ | $38.09_{\pm0.45}$ |
| +AC | $\mathbf{28.18_{\pm1.07}}$ | $\mathbf{38.75_{\pm0.26}}$ | $\mathbf{23.29_{\pm0.54}}$ | $39.83_{\pm1.22}$ | $\mathbf{50.40_{\pm0.68}}$ | $\mathbf{39.77_{\pm0.32}}$ | $\mathbf{28.96_{\pm1.13}}$ | $40.55_{\pm0.95}$ | $\mathbf{49.13_{\pm0.35}}$ | $\mathbf{48.35_{\pm0.50}}$ | $\mathbf{38.72_{\pm0.61}}$ |
| | | | | | CIFAR100x10 | | | | | | |
| Base | $10.27_{\pm0.05}$ | $34.51_{\pm0.40}$ | $22.22_{\pm0.72}$ | $30.69_{\pm0.62}$ | $42.87_{\pm1.51}$ | $32.31_{\pm0.82}$ | $10.20_{\pm0.35}$ | $29.56_{\pm0.44}$ | $38.87_{\pm0.45}$ | $42.29_{\pm0.49}$ | $29.38_{\pm0.26}$ |
| +LP | $\mathbf{17.77_{\pm1.30}}$ | $35.62_{\pm0.89}$ | $25.60_{\pm0.91}$ | $\mathbf{33.72_{\pm1.38}}$ | $44.74_{\pm2.31}$ | $35.78_{\pm0.46}$ | $18.84_{\pm0.19}$ | $\mathbf{31.88_{\pm1.11}}$ | $43.37_{\pm0.31}$ | $43.33_{\pm0.08}$ | $33.06_{\pm0.17}$ |
| +AC | $16.88_{\pm1.08}$ | $\mathbf{36.97_{\pm0.39}}$ | $\mathbf{27.74_{\pm0.73}}$ | $31.37_{\pm0.94}$ | $\mathbf{46.19_{\pm1.47}}$ | $\mathbf{37.32_{\pm0.28}}$ | $\mathbf{19.12_{\pm0.88}}$ | $30.31_{\pm1.14}$ | $\mathbf{45.67_{\pm0.52}}$ | $\mathbf{44.17_{\pm0.28}}$ | $\mathbf{33.57_{\pm0.22}}$ |

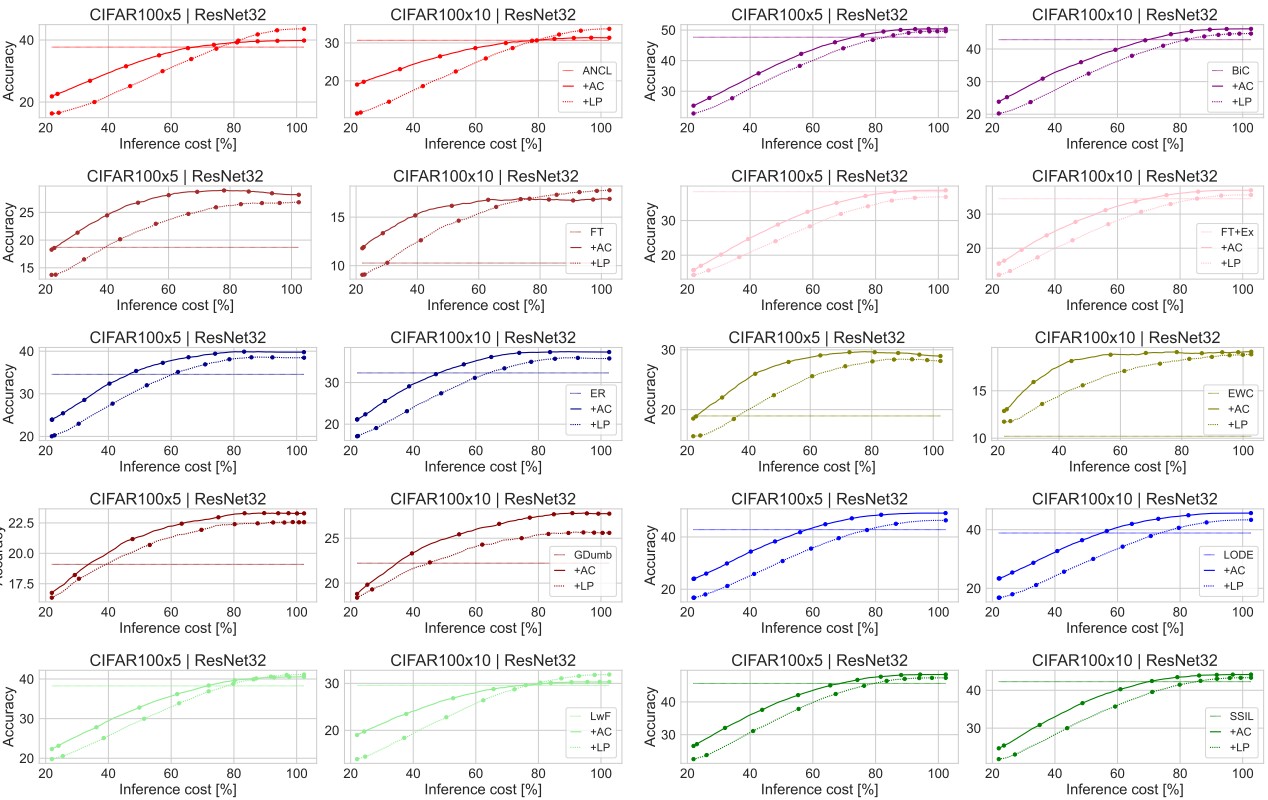

*Figure 13.* Dynamic inference plots for several continual learning methods extended with auxiliary classifiers when using auxiliary classifiers with enabled gradient propagation (AC) or without (LP).

## D.4. Leave-one-AC ablation

In this section, we perform leave-one-out ablation of the setting explored in Section 5.1 for CIFAR100 on methods explored in Section 3. Namely, we train the model with only one auxiliary classifier out of the original six, with the classifier weight equal to 1. during training and present dynamic inference results in Figure 14. For non-naive methods, we observe that either the 5th or the 6th AC achieves the best performance. Interestingly for finetuning, later ACs yield lower performance, which is consistent with our observations on more native stability in early layers. All tested AC setups achieve comparable performance at the full computational budget, but compared to our based setup of using all 6 ACs they tend to underperform at a lower compute budget. Slightly better performance of single AC setup for FT+Ex and LwF hints that AC placement in our work could be further optimized. However, overall similar performance across all the tested scenarios prove the robustness of our idea.

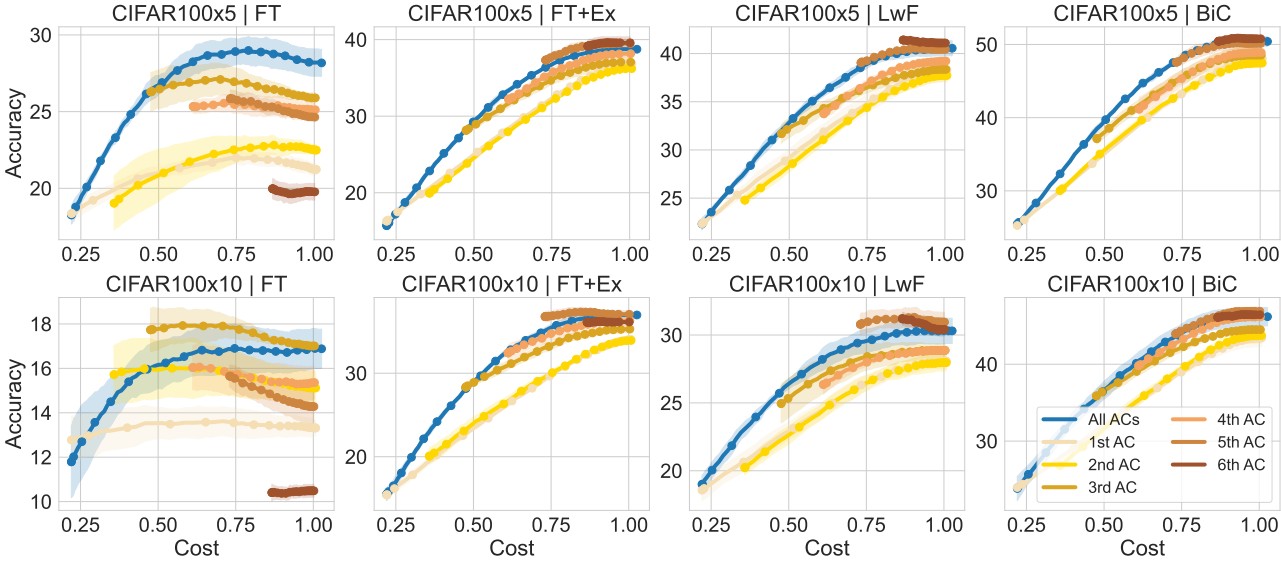

*Figure 14.* Leave one out AC ablation for FT, FT+Ex, LwF and BiC.

# E. Additional dynamic inference results

This section presents dynamic inference plots for experiments performed in Section 5 or complementary to those experiments.

## E.1. Standard benchmarks

In Figure 6, we present dynamic inference results for 5 and 10 task splits of CIFAR100 and ImageNet100 with the methods omitted in the main section due to space constraints. As in the previous sections, ACs demonstrate robustness and allow for a significant reduction of average computation.

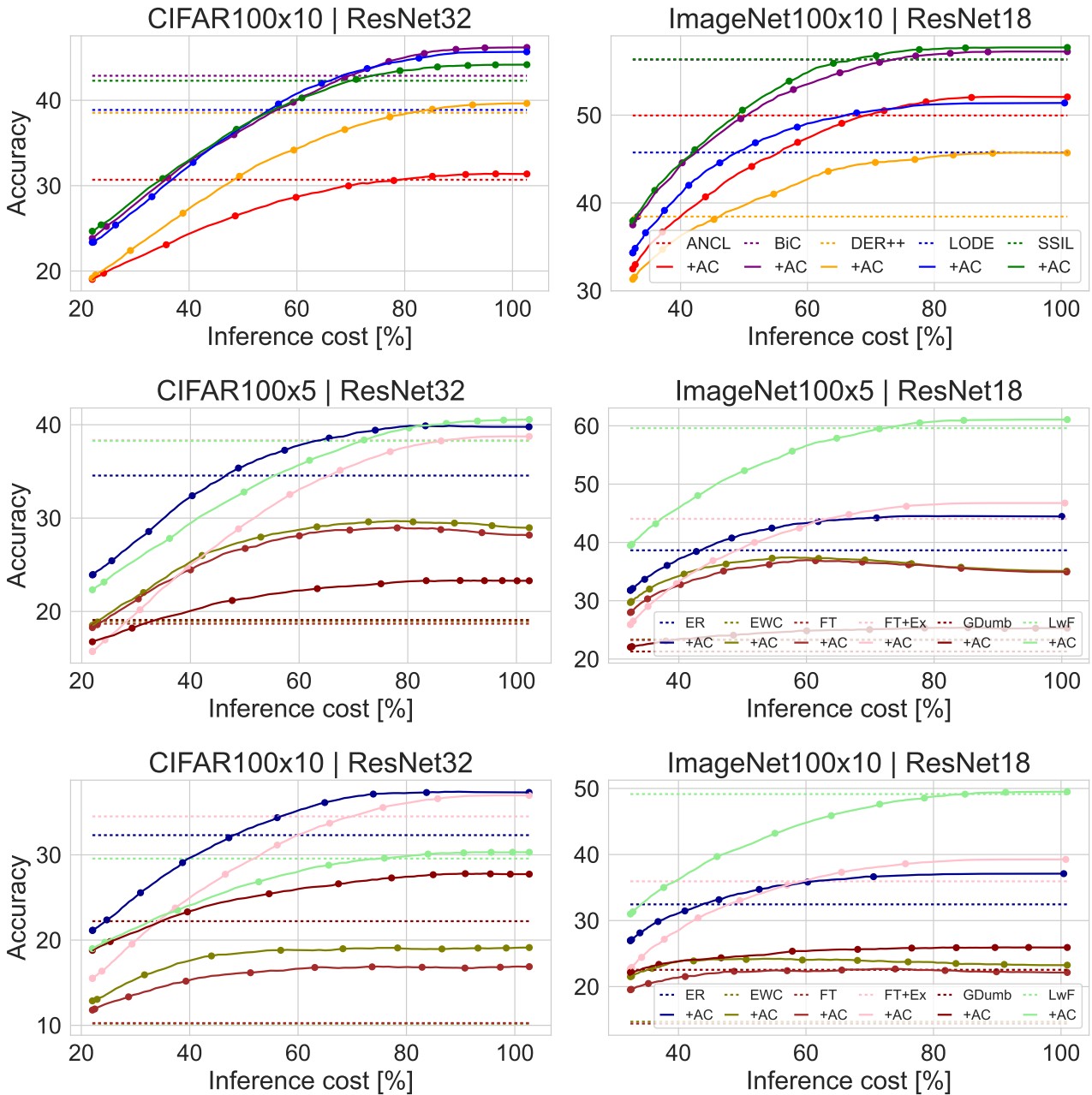

*Figure 15.* Dynamic inference plots as in Figure 6 for additional continual learning methods extended with auxiliary classifiers on CIFAR100 and ImageNet100 split into 5 and 10 tasks.

### E.2. VGG19

In Figure 7 in this section, we present dynamic inference plots for VGG19 on CIFAR100 with all the methods considered in the main paper that correspond to the results from Table 2. AC-enhanced methods outperform the baselines by even bigger margins than ResNets used in our main analysis and match their accuracy at a smaller fraction of the compute.

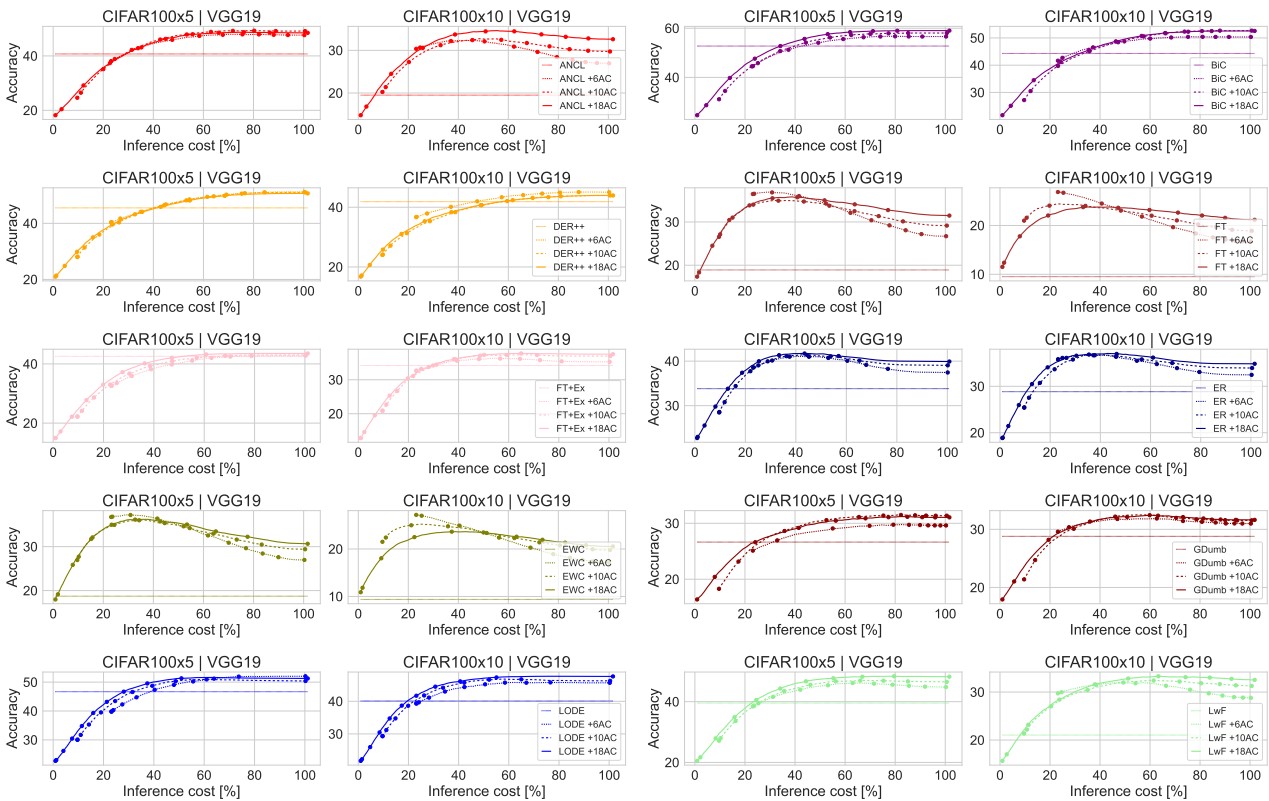

*Figure 16.* Dynamic inference experiments on CIFAR100 with VGG19 network enhanced with different number of ACs.

## E.3. ViT

In Figure 17, we show dynamic inference results with all AC-enhanced methods for ViT trained on ImageNet100. Due to computational constraints, contrary to the main experiments, we only use a single seed for these experiments. While the overall performance of ViT is below the results we achieve with ResNet18 models due to issues with training ViTs, already discussed in the paper, for all methods, AC versions outperform the baseline. Interestingly, for certain methods, performance peaks at the middle computational budgets and starts to degrade a bit, indicating insufficient calibration of Transformers trained with those methods.

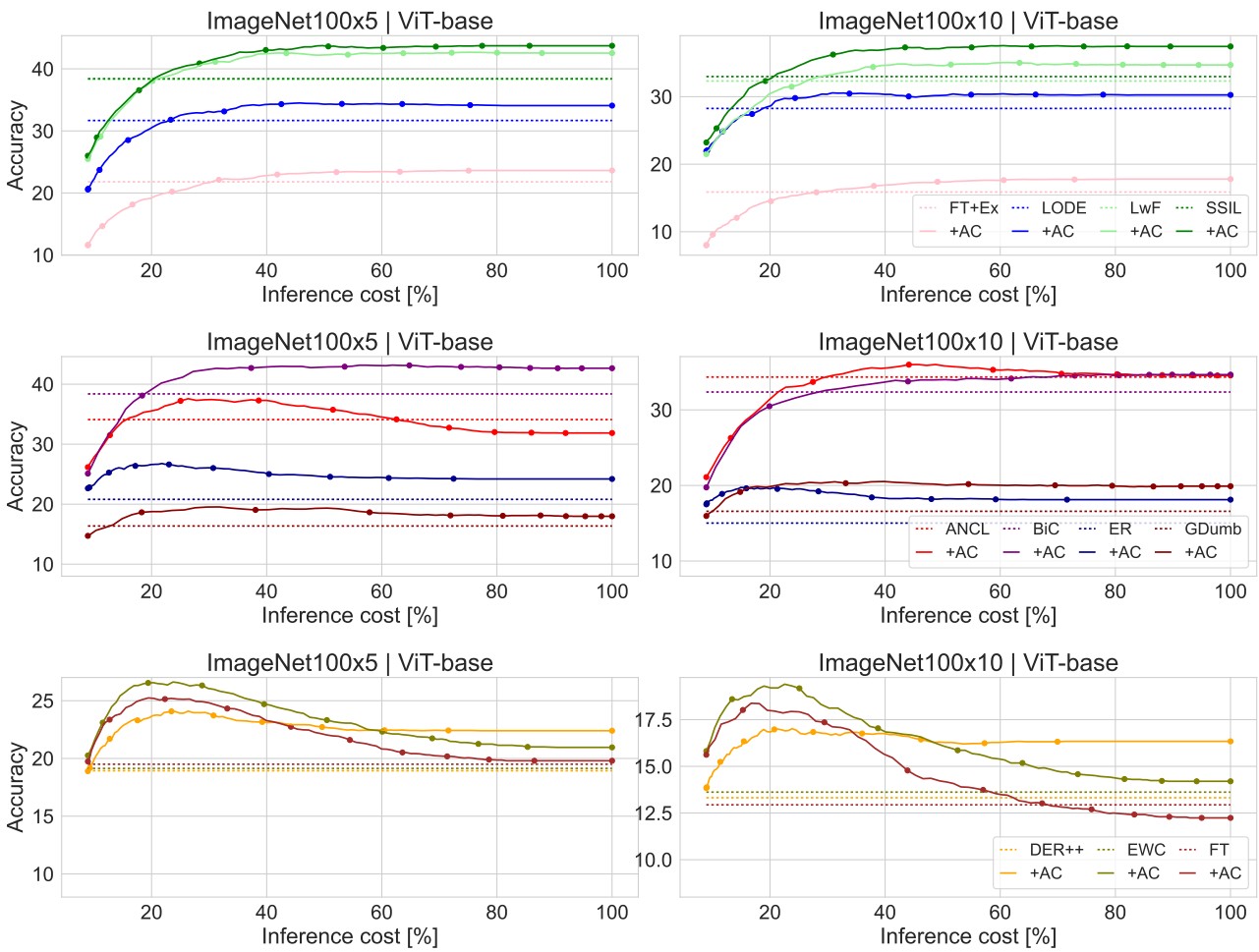

*Figure 17.* Dynamic inference plots for ViT-base on ImageNet100.

### E.4. WideResNet-2

In this section, we provide dynamic inference plots for WideResNet-2 for continual learning methods missing from Figure 8 in the main paper.

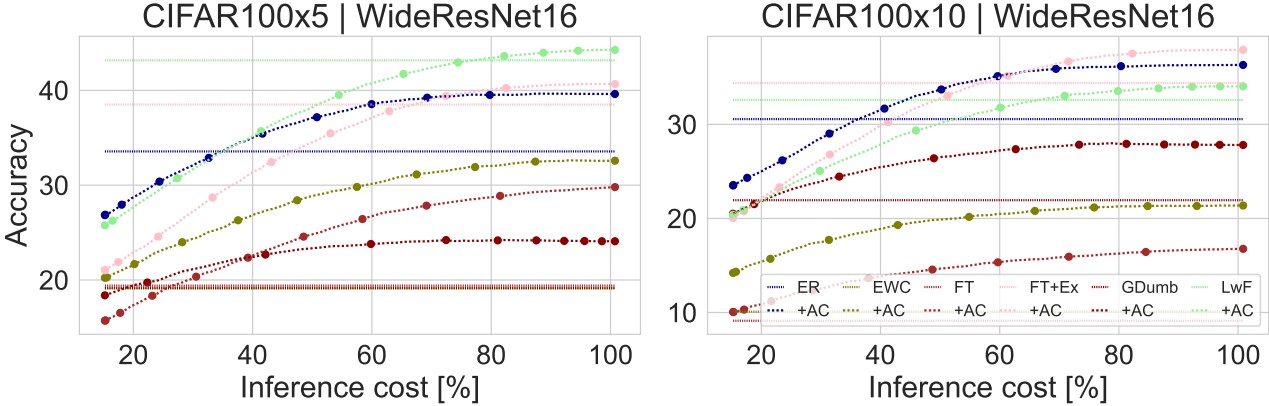

*Figure 18.* Dynamic inference plots for WideResNet-2 on CIFAR100.

## F. Additional analysis

In this section, we present additional analysis of multi-classifier networks. Furthermore, we provide results complementary to the experiments from Section 3 for 5 task split of CIFAR100, as well as the results of those experiments for the models with enabled gradient propagation.

### F.1. Forgetting in AC-enhanced models

To better understand the impact of the addition of ACs, we further analyse how they affect the network forgetting. We define average forgetting across training on total T tasks as $\frac{1}{T-1}\sum_{t=1}^{T-1}(Acc_t^t - Acc_t^T)$, where $Acc_i^j$ refers to the accuracy on the $i$-th task after training on $j$-th task. The final task is excluded from the average, as it has zero forgetting under this definition. In Table 10, we report average forgetting values at the end of training for the main settings we consider in our experimental section (ResNet32 on CIFAR100 and ResNet18 on ImageNet100). The addition of ACs generally leads to reduced forgetting. While there are isolated cases where AC-enhanced models show slightly higher forgetting, these are exceptions, and even in those cases the final accuracy is still higher with ACs, as evidenced in Table 1. These results indicate that ACs enable more effective knowledge accumulation during continual learning.

*Table 10.* Forgetting values (difference between average accuracy at the end of the first task and at the end of the training) corresponding to the experiments in Table 1. AC-enhanced models generally exhibit lower forgetting, and even in cases where the forgetting is higher, the ACs still lead to better final accuracy. This indicates that the addition of the ACs improves the network's ability to accumulate knowledge.

| Method | FT | FT+Ex | GDumb | ANCL | BiC | DER++ | ER | EWC | LwF | LODE | SSIL | Avg |
|---|---|---|---|---|---|---|---|---|---|---|---|---|
| | | | | | CIFAR100x5 | | | | | | | |
| Base | 64.85±1.27 | **44.80±0.79** | 12.01±0.63 | 36.94±0.79 | 9.33±2.89 | 7.42±4.17 | 47.25±0.36 | 64.65±1.13 | 24.21±2.69 | **20.61±1.14** | 24.80±1.08 | 32.44±19.83 |
| +AC | 50.04±1.18 | 45.51±0.66 | 10.00±0.47 | 26.32±1.15 | 9.26±4.00 | 3.68±4.05 | 40.31±0.84 | 48.04±1.34 | 19.95±0.57 | 20.61±0.96 | 19.95±1.12 | **26.70±15.90** |
| | | | | | CIFAR100x10 | | | | | | | |
| Base | 70.35±2.42 | 49.57±1.25 | 14.11±1.08 | 40.49±1.55 | 9.31±2.43 | 11.88±4.27 | 51.77±1.11 | 68.40±2.74 | **24.26±2.12** | **18.49±1.50** | 21.44±2.74 | 34.55±21.51 |
| +AC | 56.40±2.97 | 48.03±1.05 | 10.68±1.44 | 30.60±1.46 | 6.37±3.81 | 8.39±5.10 | 46.62±1.33 | 54.08±2.30 | 29.12±1.01 | 20.24±2.12 | 16.71±1.96 | **29.75±17.96** |
| | | | | | ImageNet100x5 | | | | | | | |
| Base | 74.35±0.63 | 52.00±0.75 | 16.73±0.82 | **9.41±1.06** | 10.37±1.70 | 42.74±7.04 | 56.14±1.27 | 74.49±0.97 | 18.90±0.73 | 36.32±0.98 | 21.90±0.88 | 37.58±23.08 |
| +AC | 61.45±1.30 | 49.97±1.00 | 15.98±1.07 | 13.70±0.99 | 9.76±1.05 | 14.45±2.39 | 49.65±1.25 | 61.77±0.61 | 17.49±0.76 | 31.83±2.64 | 19.68±0.80 | **31.43±19.40** |
| | | | | | ImageNet100x10 | | | | | | | |
| Base | 77.50±1.54 | 57.50±1.40 | 19.67±1.37 | **25.27±1.07** | 12.73±3.38 | 28.54±5.96 | 60.38±1.42 | 76.70±1.40 | 23.76±1.49 | 34.37±2.08 | 19.75±1.69 | 39.65±22.73 |
| +AC | 68.44±1.54 | 54.67±1.73 | 18.12±1.47 | 27.44±1.14 | 12.84±2.71 | 28.38±5.63 | 56.06±1.74 | 68.13±1.18 | 18.17±0.90 | 31.15±2.20 | 18.90±1.80 | **36.57±20.12** |

## F.2. Accuracies across the training on CIFAR100 for separate tasks

In this section, we provide more detailed results for our experiments on CIFAR100, where we compare the performance of base and AC-enhanced networks with different continual learning methods on a per-task basis. The results for each method are shown in Figures 19 to 29.

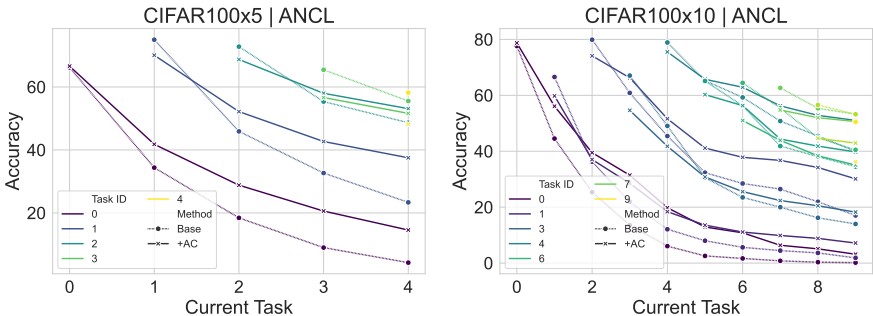

*Figure 19.* Per-task accuracy on CIFAR100x5 (left) and CIFAR100x10 (right) for ANCL.

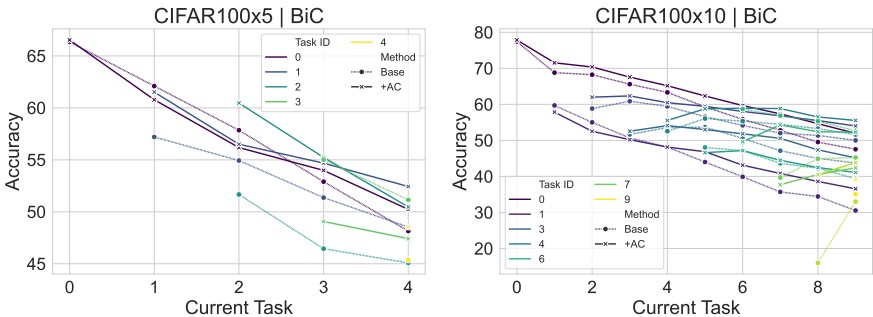

*Figure 20.* Per-task accuracy on CIFAR100x5 (left) and CIFAR100x10 (right) for BiC.

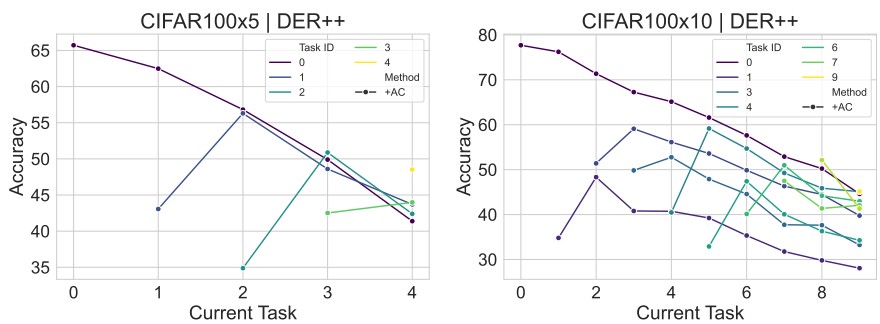

*Figure 21.* Per-task accuracy on CIFAR100x5 (left) and CIFAR100x10 (right) for DER++.

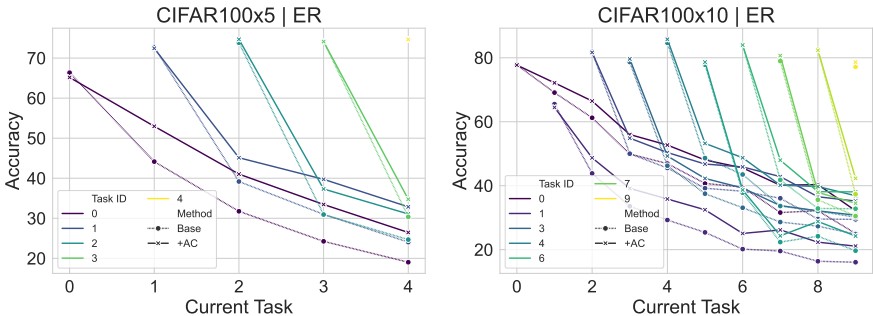

*Figure 22.* Per-task accuracy on CIFAR100x5 (left) and CIFAR100x10 (right) for ER.

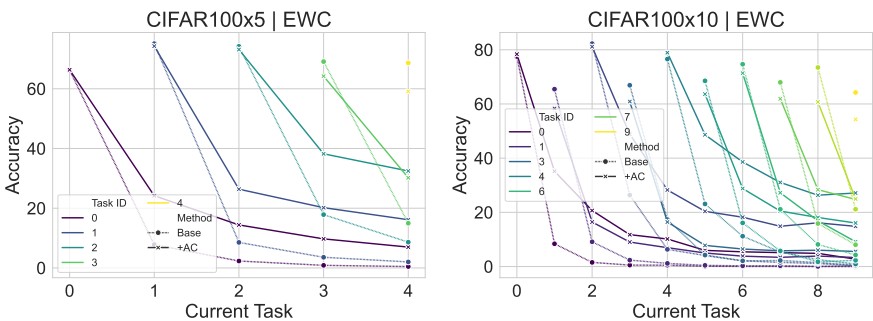

*Figure 23.* Per-task accuracy on CIFAR100x5 (left) and CIFAR100x10 (right) for EWC.

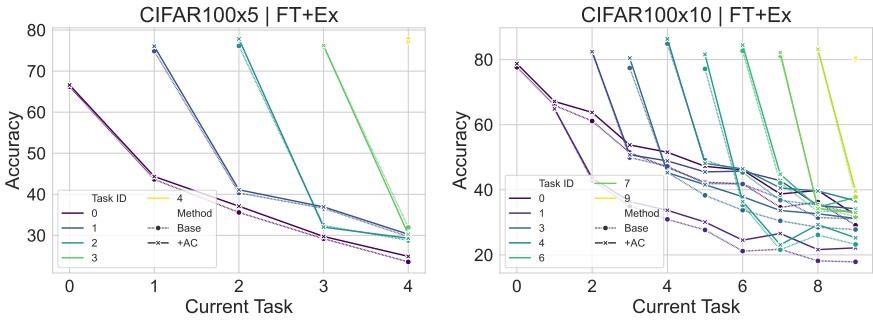

*Figure 24.* Per-task accuracy on CIFAR100x5 (left) and CIFAR100x10 (right) for FT+Ex.

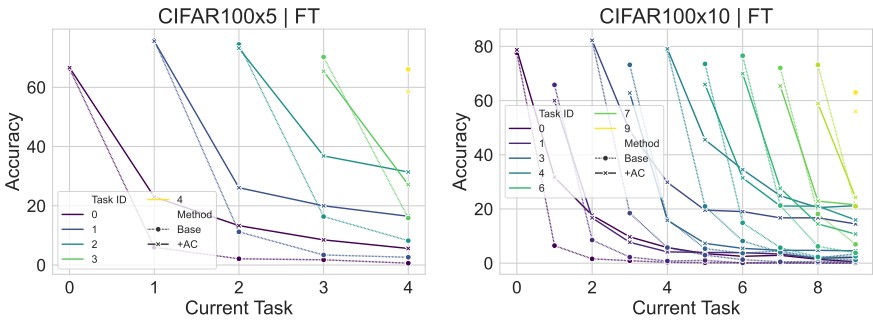

*Figure 25.* Per-task accuracy on CIFAR100x5 (left) and CIFAR100x10 (right) for FT.

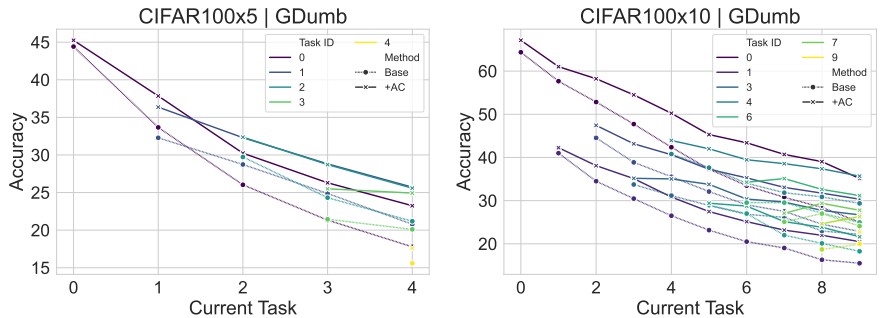

*Figure 26.* Per-task accuracy on CIFAR100x5 (left) and CIFAR100x10 (right) for GDumb.

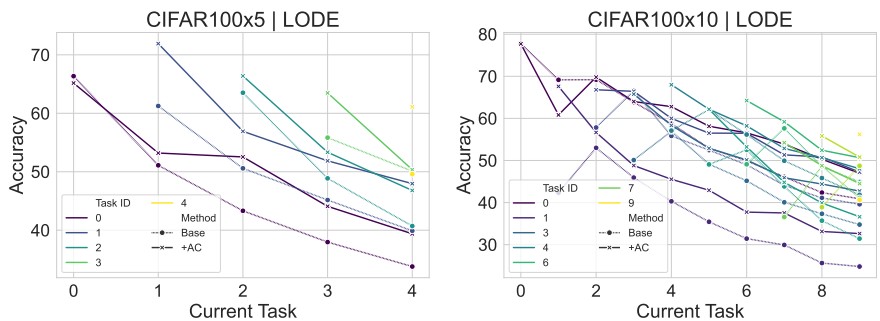

*Figure 27.* Per-task accuracy on CIFAR100x5 (left) and CIFAR100x10 (right) for LODE.

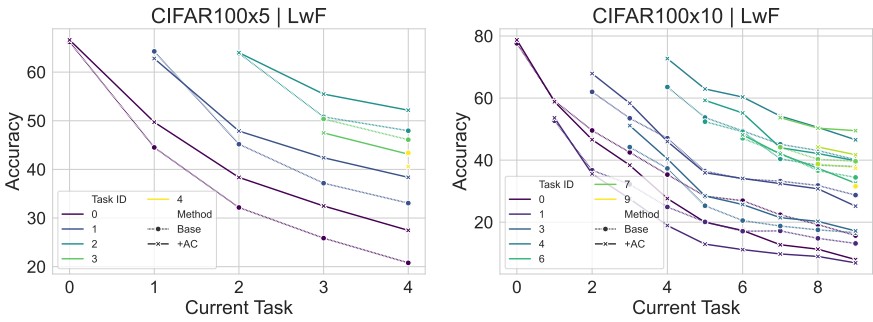

*Figure 28.* Per-task accuracy on CIFAR100x5 (left) and CIFAR100x10 (right) for LwF.

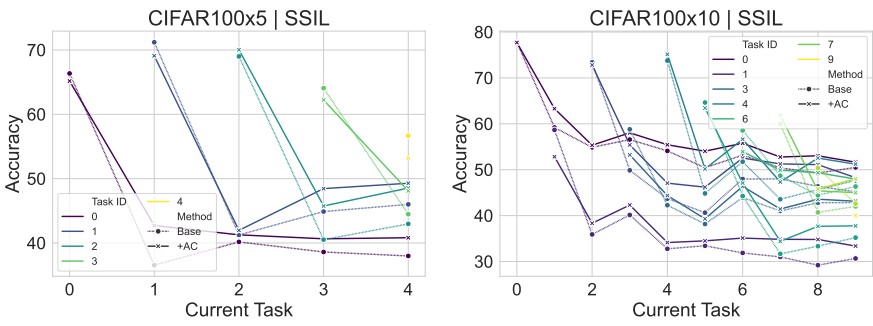

*Figure 29.* Per-task accuracy on CIFAR100x5 (left) and CIFAR100x10 (right) for SSIL.

## F.3. Results complementary to Section 3

### F.3.1. CKA

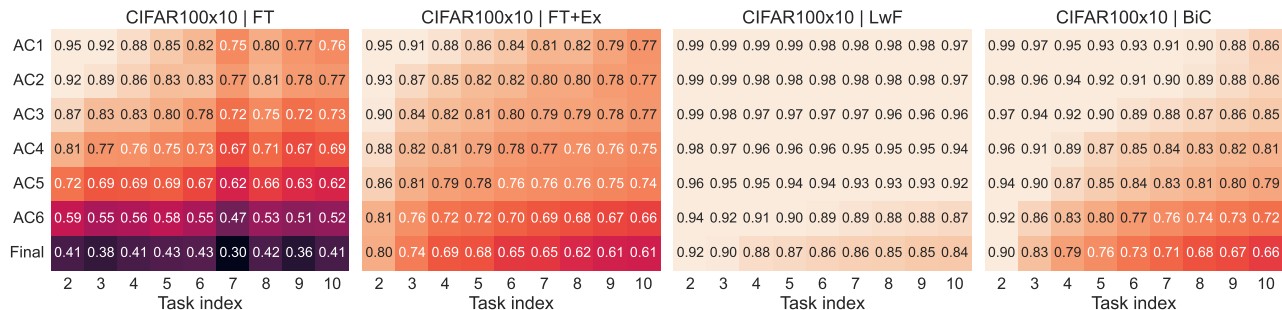

*Figure 30.* CKA of the first task representations across ResNet32 layers through continual learning on CIFAR100 split into 5 tasks. ACs are trained without gradient propagation.

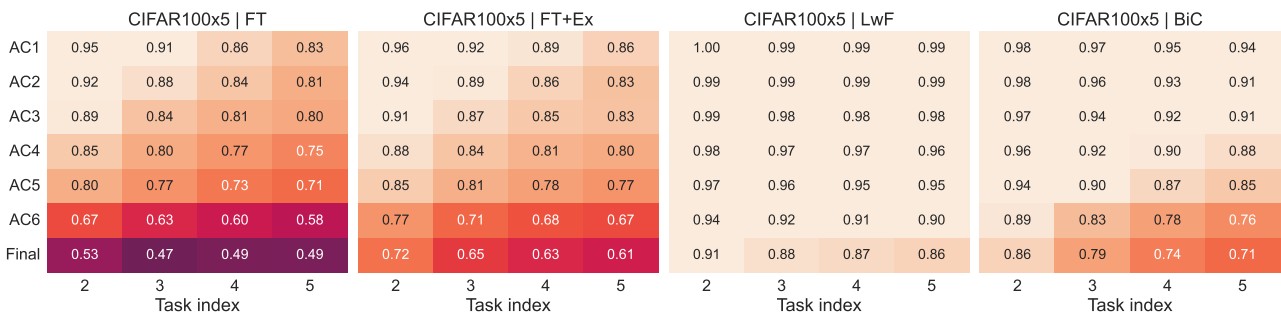

*Figure 31.* CKA of the first task representations across ResNet32 layers through continual learning on CIFAR100 split into 10 tasks. ACs are trained **with** gradient propagation enabled.

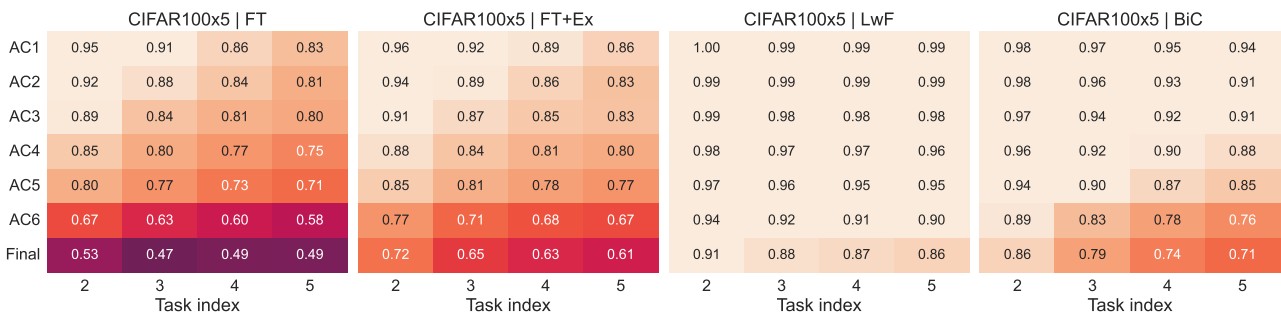

*Figure 32.* CKA of the first task representations across ResNet32 layers through continual learning on CIFAR100 split into 5 tasks. ACs are trained **with** gradient propagation enabled.

### F.3.2. AC PERFORMANCE DIFFERENCE OVER THE FINAL CLASSIFIER

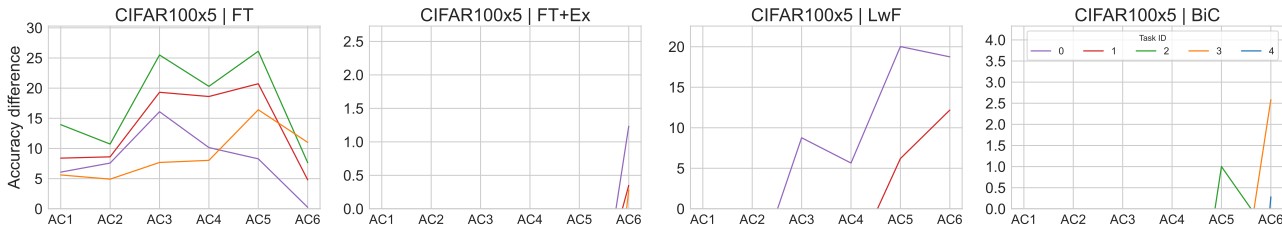

*Figure 33.* Per-task **difference** between the accuracy (only positives) of the ACs trained with linear probing for 5 task split of CIFAR100.

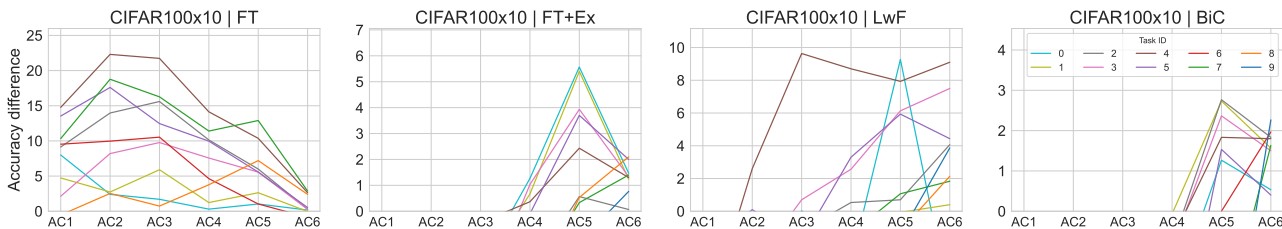

*Figure 34.* Per-task **difference** between the accuracy (only positives) of the ACs trained with **gradient propagation** for 10 task split of CIFAR100.

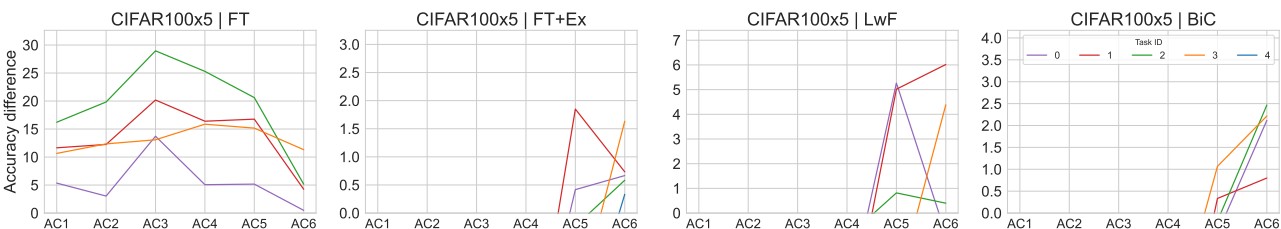

*Figure 35.* Per-task **difference** between the accuracy (only positives) of the ACs trained with **gradient propagation** for 5 task split of CIFAR100.

### F.3.3. OVERTHINKING

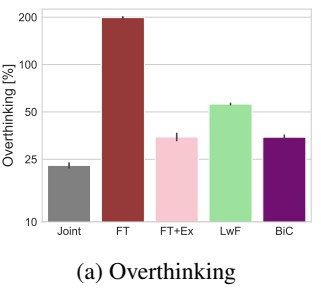

(a) Overthinking

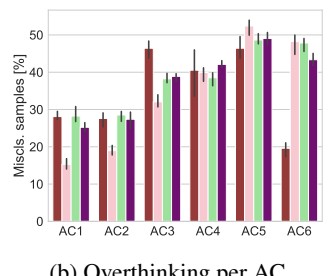

(b) Overthinking per AC

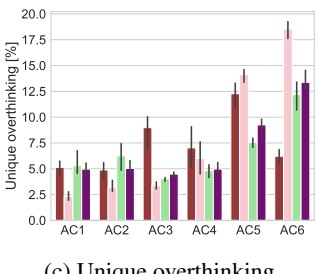

(c) Unique overthinking

*Figure 36.* Overthinking for 5 task split of CIFAR100, without gradient propagation.

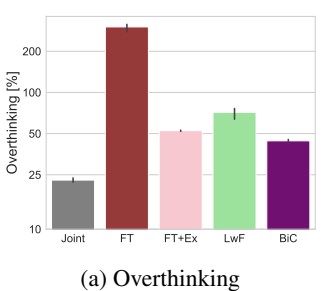

(a) Overthinking

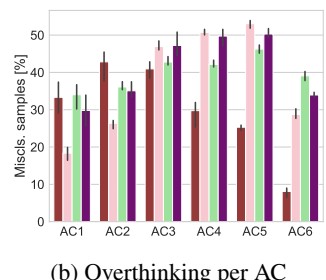

(b) Overthinking per AC

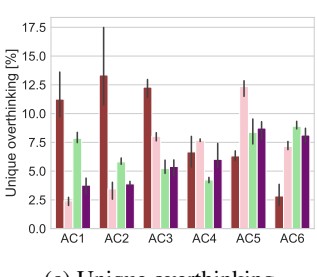

(c) Unique overthinking

*Figure 37.* Overthinking for 10 task split of CIFAR100, **with** gradient propagation enabled.

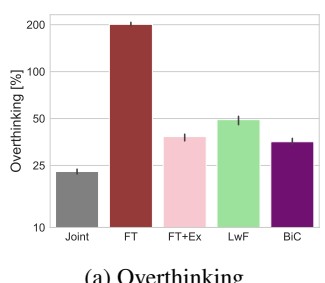

(a) Overthinking

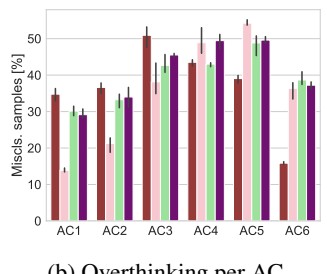

(b) Overthinking per AC

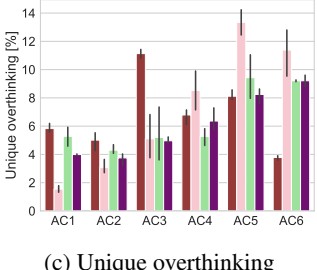

(c) Unique overthinking

*Figure 38.* Overthinking for 10 task split of CIFAR100, **with** gradient propagation enabled.

### F.3.4. TRAINING ACCURACY CHANGE FOR CIFAR100 5 TASK SPLIT

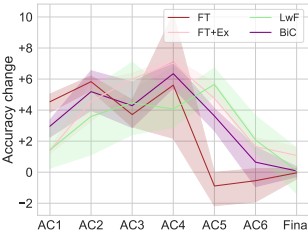

*Figure 39.* Difference between classifiers performance after enabling gradient propagation on CIFAR100 split into 5 tasks.

## F.4. Classifier selection patterns and accuracy during inference

In Figures 40 and 41, we show the classifier selection distribution and accuracy for static inference in the CIFAR100 experiments from Table 1. While early classifiers are rarely selected, intermediate classifiers often match or outperform the final classifier. We hope these results help to explain the performance improvements from adding the ACs.

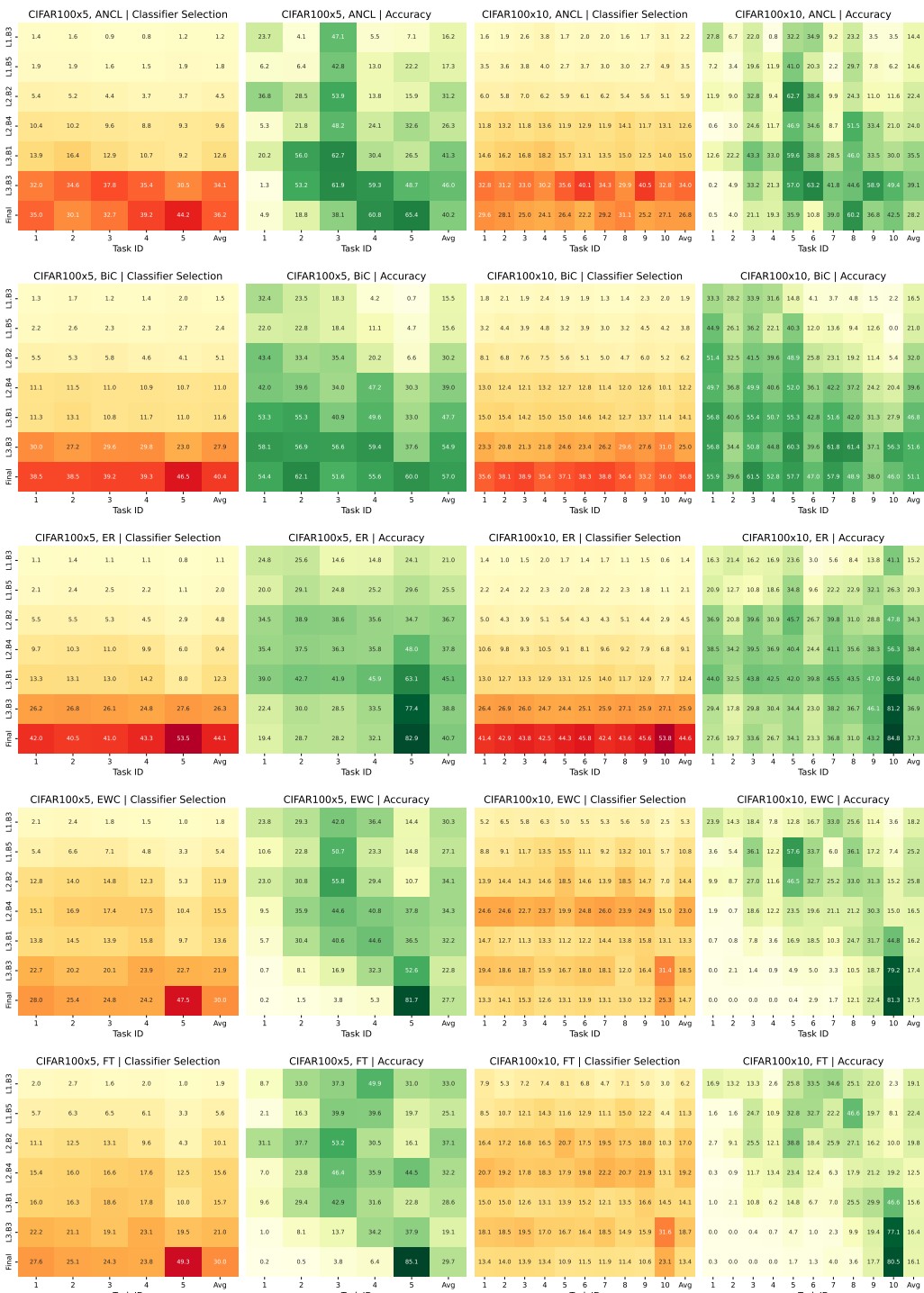

*Figure 40.* Distribution and accuracy of classifiers for ANCL, BiC, ER, EWC and FT.

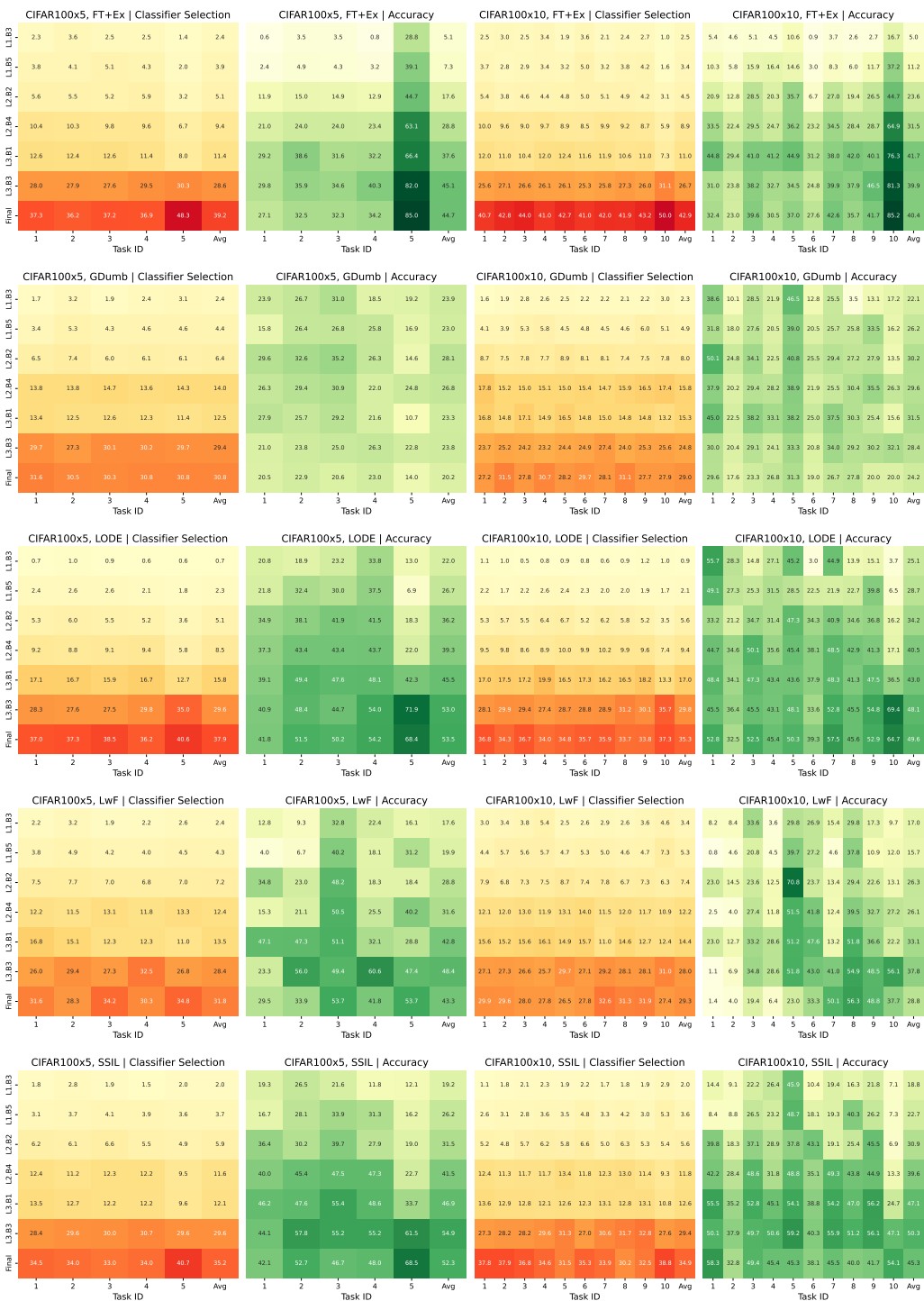

*Figure 41.* Distribution and accuracy of classifiers for FT+Ex, GDumb, LODE, LwF and SSIL.

## F.5. Diversity of the auxiliary classifier predictions

To investigate the diversity among the auxiliary classifiers, we measure *unique accuracy* - the percentage of the samples correctly predicted only by this given classifier and are misclassified by all other classifiers - and present the results in Figures 42 to 45. The intermediate classifiers learn to specialize to some degree across all analyzed methods and all intermediate classifiers, especially on the older tasks, with the trend more visible for the naive and exemplar-free settings.

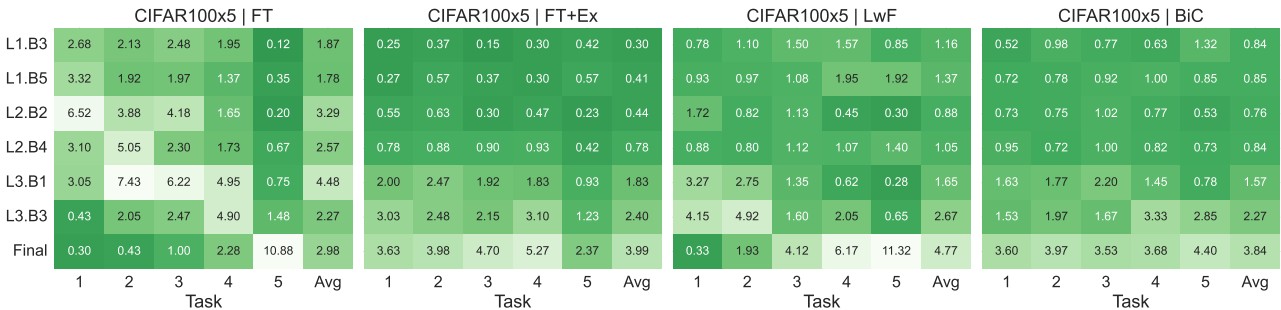

Figure 42. Percentage of unique samples that only a single given classifier classifies correctly for different tasks on CIFAR100 split into 5 tasks. The classifiers are trained without gradient propagation.

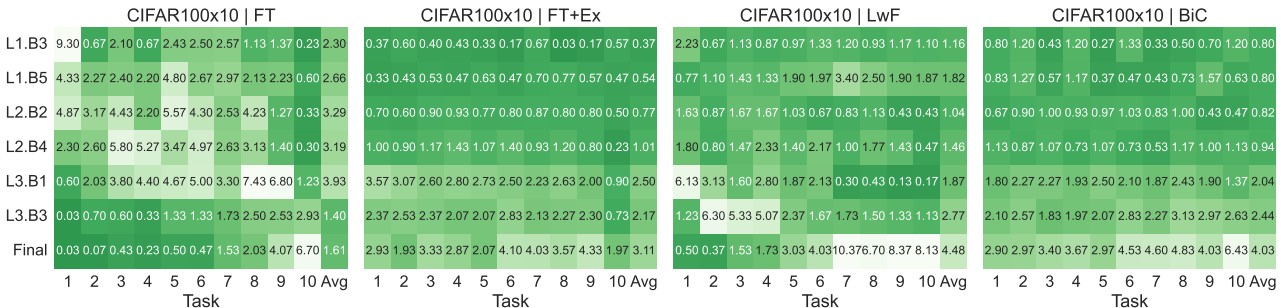

Figure 43. Percentage of unique samples that only a single given classifier classifies correctly for different tasks on CIFAR100 split into 10 tasks. The classifiers are trained without gradient propagation.

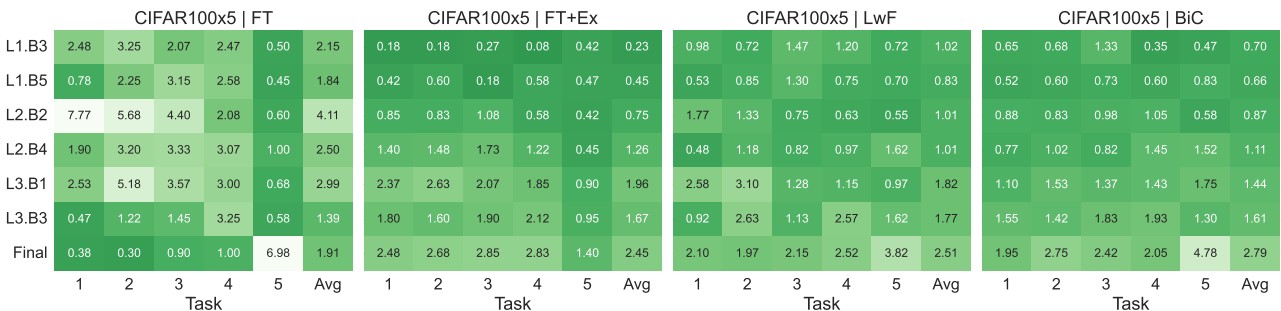

Figure 44. Percentage of unique samples that only a single given classifier classifies correctly for different tasks on CIFAR100 split into 5 tasks. The classifiers are trained **with gradient propagation enabled**.

## F.6. Upper bound analysis of multi-classifier networks

To quantify the potential of the auxiliary classifiers, we consider an oracle multi-classifier network as an upper bound for our method. When evaluating such an oracle, we obtain predictions from all of its classifiers and consider a prediction correct when at least one classifier (auxiliary classifier or the original network classifier) is correct. Therefore, the oracle has an

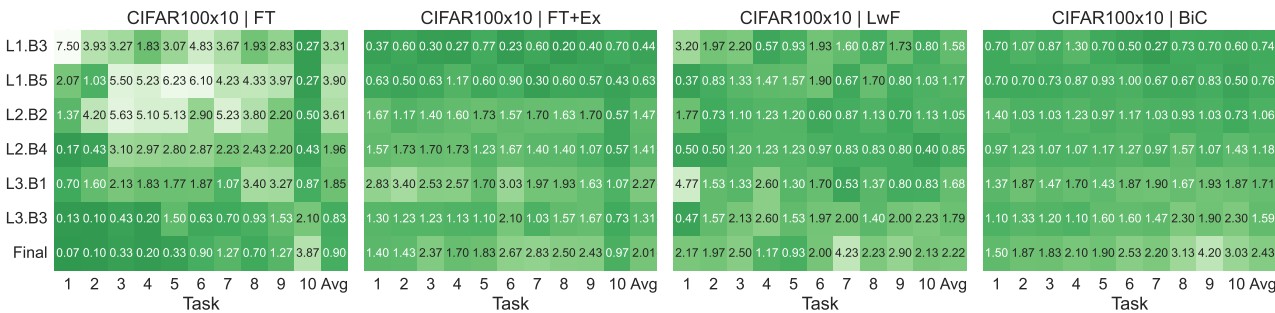

*Figure 45.* Percentage of unique samples that only a single given classifier classifies correctly for different tasks on CIFAR100 split into 10 tasks. The classifiers are trained **with gradient propagation enabled**.

ideal algorithm for combining classifier predictions and always returns the 'best case' prediction from all the classifiers. We measure the difference between the accuracy of such an oracle network and the accuracy of a standard single-classifier network and present the results for first task data, last task data, and average over all the tasks in Table 11 and Table 12 for both linear probing and ACs. As in our previous analysis in Section 3, exemplar-free methods show more variance in the performance across tasks. However, the average difference across all tasks is also significant for exemplar-based methods, with the oracle for the best-performing method - BiC - achieving approximately a 30-40% relative increase in overall accuracy. Those results indicate that, while our simple setup achieves consistent improvement, it is still leaves room for improvement in the future work.

*Table 11.* Upper bound on accuracy improvement on 5 tasks of CIFAR100 when using oracle multi-classifier network, trained with linear probing and auxiliary classifiers.

|       | Task 1 | Task 2 | Task 3 | Task 4 | Task 5 | Avg |
|-------|--------|--------|--------|--------|--------|-----|
| \multicolumn{7}{c}{CIFAR100x5, Linear Probing} | | | | | | |
| FT    | 31.82±2.23 | 46.90±1.60 | 54.22±0.59 | 43.30±1.24 | 6.57±0.96 | 36.56±0.52 |
| FT+Ex | 12.28±0.10 | 14.58±0.33 | 12.72±1.80 | 13.87±0.93 | 11.37±0.71 | 12.96±0.62 |
| LwF   | 35.07±2.03 | 28.88±2.51 | 20.92±2.01 | 15.25±1.13 | 9.55±1.47 | 21.93±0.20 |
| BiC   | 15.15±0.91 | 17.00±2.33 | 18.73±1.50 | 17.93±2.26 | 16.03±2.39 | 16.97±0.50 |
| \multicolumn{7}{c}{CIFAR100x5, Auxiliary Classifiers} | | | | | | |
| FT    | 24.65±2.82 | 45.07±5.77 | 56.87±3.68 | 48.82±4.36 | 9.05±0.65 | 36.89±1.33 |
| FT+Ex | 14.42±0.60 | 17.42±1.12 | 15.70±1.58 | 15.30±1.59 | 10.88±0.40 | 14.74±0.66 |
| LwF   | 18.52±1.89 | 22.57±2.84 | 21.57±0.93 | 18.57±0.95 | 15.07±2.08 | 19.26±0.61 |
| BiC   | 15.88±0.96 | 18.42±0.46 | 18.83±2.29 | 18.72±0.06 | 15.43±3.06 | 17.46±0.61 |

*Table 12.* Upper bound on accuracy improvement on 10 tasks of CIFAR100 when using oracle multi-classifier network, trained with linear probing and auxiliary classifiers.

|       | Task 1 | Task 2 | Task 3 | Task 4 | Task 5 | Task 6 | Task 7 | Task 8 | Task 9 | Task 10 | Avg |
|-------|--------|--------|--------|--------|--------|--------|--------|--------|--------|---------|-----|
| \multicolumn{12}{c}{CIFAR100x10, Linear Probing} | | | | | | | | | | | |
| FT    | 30.00±7.52 | 16.10±3.90 | 40.47±6.03 | 23.37±4.57 | 48.70±0.72 | 41.43±5.52 | 32.80±1.78 | 44.40±4.83 | 28.87±0.97 | 9.03±1.72 | 31.52±0.64 |
| FT+Ex | 17.17±1.24 | 16.63±0.42 | 18.40±2.72 | 18.10±1.44 | 17.67±0.78 | 16.30±1.31 | 18.90±0.79 | 15.77±1.30 | 17.00±0.44 | 11.90±1.32 | 16.78±0.38 |
| LwF   | 48.93±6.24 | 30.30±3.00 | 33.63±3.35 | 28.93±5.05 | 25.60±2.13 | 21.50±5.50 | 13.30±2.17 | 14.47±1.50 | 11.33±1.24 | 8.40±1.54 | 23.64±1.29 |
| BiC   | 19.03±1.10 | 20.73±2.04 | 20.50±2.74 | 20.83±2.37 | 17.83±1.12 | 19.17±0.67 | 16.83±3.16 | 18.90±0.53 | 20.77±4.80 | 17.73±4.31 | 19.23±0.70 |
| \multicolumn{12}{c}{CIFAR100x10, Auxiliary Classifiers} | | | | | | | | | | | |
| FT    | 12.93±0.38 | 14.30±4.69 | 35.07±0.21 | 25.70±4.90 | 46.27±4.57 | 38.70±2.77 | 32.50±2.17 | 43.77±0.93 | 34.73±3.46 | 9.20±0.62 | 29.32±0.98 |
| FT+Ex | 20.03±1.56 | 17.77±1.47 | 18.87±0.60 | 20.33±1.70 | 18.13±2.25 | 20.50±0.78 | 18.97±1.95 | 17.47±1.36 | 19.13±2.48 | 12.30±0.80 | 18.35±0.01 |
| LwF   | 22.80±3.46 | 12.60±4.18 | 22.93±1.65 | 21.50±2.91 | 28.30±1.80 | 25.33±4.65 | 12.70±1.49 | 21.53±3.31 | 18.00±5.12 | 15.27±2.77 | 20.10±0.74 |
| BiC   | 18.87±2.76 | 20.60±0.46 | 21.53±2.74 | 22.70±2.52 | 18.60±1.51 | 19.87±2.01 | 17.17±1.88 | 19.77±0.57 | 18.77±8.36 | 17.73±7.03 | 19.56±0.78 |

## F.7. Dynamic inference rule ablation

In Table 13 we demonstrate the accuracy of two variants of dynamic inference for different confidence thresholds $\lambda$ for CIFAR100. We compare the standard, early-exit paradigm, where the network returns a final classifier prediction in case no classifier meets the exit rule, and the paradigm used in our experiments where the network defaults to the most confident prediction. Using the most confident prediction outperforms the standard early-exit rule, which is consistent with our analysis that showed that the last classifier is not always the best in continual learning and that the early classifiers exhibit lower forgetting for earlier task data.

*Table 13.* Comparison between dynamic inference performance with different confidence thresholds $\lambda$ when using maximum confidence prediction (MC) and final classifier prediction (Last) as the default output for multi-classifier networks trained with linear probing (LP) or jointly with the network with gradient propagation (AC). Using max confidence prediction yields better accuracy.

| | $\lambda$ | FT (LP) | FT+Ex (LP) | LwF (LP) | BiC (LP) | FT (AC) | FT+Ex (AC) | LwF (AC) | BiC (AC) |
|---|---|---|---|---|---|---|---|---|---|
| | | | | | CIFAR100x5 | | | | |
| Last | 0.5 | 24.14±1.35 | 28.43±0.87 | 37.34±0.09 | 48.35±0.23 | 27.64±0.97 | 28.87±0.32 | 38.08±0.87 | 48.15±0.40 |
| MC | | 24.73±1.48 | 28.33±0.96 | 36.95±0.20 | 48.48±0.40 | 28.10±1.03 | 28.85±0.31 | 38.36±0.59 | 48.37±0.31 |
| Last | 0.75 | 23.67±1.32 | 35.60±1.26 | 40.21±0.08 | 49.25±0.33 | 26.00±0.67 | 36.27±0.28 | 39.41±1.01 | 49.45±0.73 |
| MC | | 26.66±1.70 | 35.09±1.38 | 39.70±0.10 | 49.74±0.48 | 28.91±1.07 | 36.18±0.17 | 40.33±0.76 | 50.19±0.63 |
| Last | 0.9 | 20.98±0.99 | 37.27±1.10 | 39.97±0.27 | 49.18±0.36 | 22.38±0.33 | 38.22±0.19 | 39.28±1.28 | 49.35±0.65 |
| MC | | 26.70±1.55 | 36.57±1.33 | 40.07±0.10 | 49.89±0.52 | 28.44±1.04 | 38.27±0.13 | 40.50±0.91 | 50.39±0.65 |
| Last | 0.95 | 19.91±0.76 | 37.48±0.98 | 39.62±0.23 | 49.15±0.31 | 20.69±0.24 | 38.50±0.17 | 39.25±1.36 | 49.24±0.63 |
| MC | | 26.74±1.40 | 36.77±1.29 | 40.07±0.08 | 49.89±0.52 | 28.25±1.05 | 38.62±0.21 | 40.53±0.94 | 50.40±0.66 |
| Last | 0.98 | 19.19±0.27 | 37.46±0.92 | 39.40±0.21 | 49.14±0.32 | 19.49±0.18 | 38.55±0.38 | 39.22±1.34 | 49.24±0.65 |
| MC | | 26.83±1.23 | 36.81±1.27 | 40.10±0.07 | 49.89±0.52 | 28.19±1.08 | 38.72±0.24 | 40.55±0.95 | 50.40±0.68 |
| Last | 0.99 | 18.94±0.15 | 37.44±0.88 | 39.32±0.22 | 49.14±0.32 | 19.08±0.20 | 38.55±0.44 | 39.22±1.35 | 49.23±0.64 |
| MC | | 26.82±1.22 | 36.83±1.27 | 40.10±0.07 | 49.89±0.52 | 28.20±1.09 | 38.75±0.27 | 40.55±0.95 | 50.40±0.68 |
| Last | 1.0 | 18.39±0.08 | 37.43±0.85 | 39.11±0.26 | 49.14±0.32 | 18.35±0.30 | 38.51±0.43 | 39.22±1.34 | 49.22±0.65 |
| MC | | 26.82±1.19 | 36.83±1.27 | 40.10±0.07 | 49.89±0.52 | 28.18±1.07 | 38.75±0.26 | 40.55±0.95 | 50.40±0.68 |
| | | | | | CIFAR100x10 | | | | |
| Last | 0.5 | 15.22±1.64 | 27.03±0.90 | 28.69±0.79 | 42.37±1.60 | 15.85±1.10 | 27.68±0.42 | 28.96±1.29 | 42.40±1.17 |
| MC | | 16.05±1.95 | 27.04±0.87 | 28.06±0.93 | 42.62±1.75 | 16.79±1.34 | 27.72±0.37 | 29.09±1.10 | 42.67±1.44 |
| Last | 0.75 | 14.12±0.98 | 33.69±1.20 | 31.10±0.87 | 43.95±1.69 | 13.31±1.17 | 34.40±0.47 | 28.65±1.72 | 44.93±0.93 |
| MC | | 17.47±1.51 | 33.84±1.03 | 30.17±0.77 | 44.59±1.75 | 16.77±1.22 | 34.64±0.41 | 29.72±1.19 | 45.83±1.51 |
| Last | 0.9 | 12.19±0.51 | 34.61±1.05 | 31.03±0.92 | 43.87±1.63 | 11.25±1.04 | 35.83±0.51 | 28.30±1.83 | 44.55±0.62 |
| MC | | 17.71±1.33 | 35.41±0.87 | 30.54±0.76 | 44.82±1.71 | 16.82±1.14 | 36.59±0.52 | 29.79±1.21 | 46.14±1.46 |
| Last | 0.95 | 11.25±0.55 | 34.49±1.10 | 30.63±1.10 | 43.79±1.72 | 10.62±0.83 | 35.63±0.43 | 28.27±1.79 | 44.33±0.53 |
| MC | | 17.74±1.31 | 35.58±0.92 | 30.59±0.67 | 44.84±1.72 | 16.89±1.11 | 36.92±0.39 | 29.79±1.21 | 46.17±1.45 |
| Last | 0.98 | 10.51±0.35 | 34.27±1.13 | 30.17±1.19 | 43.77±1.72 | 10.09±0.73 | 35.29±0.39 | 28.22±1.81 | 44.23±0.54 |
| MC | | 17.77±1.30 | 35.61±0.90 | 30.60±0.68 | 44.84±1.73 | 16.88±1.08 | 36.97±0.40 | 29.79±1.21 | 46.19±1.47 |
| Last | 0.99 | 10.19±0.37 | 34.26±1.10 | 30.00±1.20 | 43.76±1.72 | 9.92±0.69 | 35.10±0.45 | 28.20±1.83 | 44.21±0.53 |
| MC | | 17.77±1.30 | 35.61±0.89 | 30.60±0.68 | 44.84±1.73 | 16.88±1.08 | 36.98±0.40 | 29.79±1.21 | 46.19±1.47 |
| Last | 1.0 | 9.79±0.33 | 34.22±1.08 | 29.65±1.19 | 43.75±1.72 | 9.76±0.63 | 34.93±0.40 | 28.19±1.84 | 44.19±0.51 |
| MC | | 17.77±1.30 | 35.62±0.89 | 30.60±0.69 | 44.84±1.73 | 16.88±1.08 | 36.97±0.39 | 29.79±1.21 | 46.19±1.47 |

