# OpenReview forum: "Improving Continual Learning Performance and Efficiency with Auxiliary Classifiers"
_ICML.cc/2025/Conference — ICML 2025 poster_

### Official Review · Reviewer_urVj · 2025-02-23

**Overall Recommendation:** 3

**Summary:**

The paper proposes a new method for continual learning, which trains auxiliary classifiers on top of the intermediate features of the deep neural network along with the main neural network. The method is motivated by the observation that, in the process of class incremental continual learning, intermediate features change less drastically than the final layer(s), and some of the auxiliary classifiers can achieve better performance than the final classifier. Results show that auxiliary classifiers can improve the performance of a number of competitive continual learning approaches, including replay, regularization, and architectural methods. With dynamic layer skipping, the ACs can also enable cheaper inference cost while keeping the same continual learning performance.

**Claims And Evidence:**

Yes, the claims made in the submission are supported by clear and convincing evidence.

**Essential References Not Discussed:**

The related works are sufficiently discussed in the paper.

**Experimental Designs Or Analyses:**

The experimental designs and analysis in the paper are sound and valid. The main experiment shown in table 1 demonstrate the effectiveness of the auxiliary classifiers across a range of different CL methods. The additional experiments shows in table 2 and figure 7 demonstrate the effectiveness of the auxiliary classifiers across different neural network architectures.

**Methods And Evaluation Criteria:**

Yes, the proposed methods and/or evaluation criteria make sense for the problem at hand. CIFAR 100x10 and ImageNet 100x10 are popular and standard evaluation benchmarks for class incremental continual learning in computer vision and classification. The baseline methods that the authors choose are good representatives of different types of continual learning algorithms.

**Other Comments Or Suggestions:**

1. The Riemer et al., 2018 paper is referred to as "ER" in the paper, which is a little bit confusing. I believe "ER" usually refers to naive Experience Replay (the FT+Ex setting in this paper), while the Riemer et al., 2018 paper is referred to as MER (Meta Experience Replay). In the provided anonymous code base the "er.py" file also seems to only contain code for naive experience replay. I would like the authors to confirm that the "ER" columns in the tables actually refer to the method proposed in the Riemer et al., 2018 paper.
2. I have some difficulty understanding what figure 4a is plotting exactly ("we plot overthinking normalized by the accuracy of the final classifier" seems a bit unclear). I would like the authors to more clearly define how the numbers plotted in Figure 4a is calculated.

**Other Strengths And Weaknesses:**

Strengths;
- The proposed method is well motivated, and draws inspirations from prior works in continual learning, neural network overthinking, and early-exit classifiers. It provides a natural tradeoff between final accuracy and inference cost.
- The analysis on overthinking of continually-learned models is a novel contribution.

Weaknesses:
- If we just care about the optimal final accuracy (the point at 100% inference cost), the amount of improvement provided by auxiliary classifiers seems limited for more recent methods, especially considering the additional training overhead of these auxiliary classifiers.
- (minor) The observation that intermediate features are more stable in continual learning methods is not a novel one, as mentioned in the paper in lines 186-189.

**Questions For Authors:**

1. In figure 2, the earlier layers (such as AC1) changes least during the process of continual learning, while AC6 changes most (closet to the final layer). However, in figure 3, we notice that AC6 provides most performance improvement for a majority of the tasks in non-trivial CL methods. Is there an intuition on why this is the case? This seems to contradict with the motivation provided in section 3.1 that the intermediate feature are more useful for classification in CL because they are more stable.
2. Is there any justification on using the confidence score to choose which auxiliary classifier to use for the final prediction (instead of say a popular vote of all the auxiliary classifiers), especially given that neural networks are often not very well calibrated?

**Relation To Broader Scientific Literature:**

The proposed method is highly related to previous works on overthinking in neural networks and early-exit classifiers for cheaper inference cost in neural networks. This paper studies these topic in the context of continual learning. The observations that overthinking is more apparent in continual learning motivates the auxiliary classifier method. The early-exit classifiers motivates the dynamic layer skipping approach proposed in the paper for inference cost - accuracy tradeoff.

The paper is also highly related to previous works on using intermediate features in continual learning. The prior finding that intermediate features are more stable motivates the auxiliary classifier method.

**Theoretical Claims:**

N/A. The paper does not include any proofs for theoretical claims in the main text.

---

> ### Author Rebuttal · Authors · 2025-03-30
>
> We thank the Reviewer for the time spent on our work and the suggestions for improving its quality. Below, we address the points raised by the Reviewer.
>
> > If we just care about the optimal final accuracy (the point at 100% inference cost), the amount of improvement provided by auxiliary classifiers seems limited for more recent methods, especially considering the additional training overhead of these auxiliary classifiers.
>
> We do not consider ACs a standalone method, but rather a addon-method that enhances existing approaches. Our experiments demonstrate that ACs enable inference cost savings up to while providing moderate performance improvements, even for newer methods such as LODE, where we both improve the performance and save up to 20% computation without any performance degradation, or save 50% compute while keeping the same performance as the baseline (see Figure 1).
>
> > The observation that intermediate features are more stable in continual learning methods is not a novel one, as mentioned in the paper in lines 186-189.
>
> As the Reviewer notes, we acknowledge prior work on the stability of intermediate features and we treat it as a motivation for our approach. To our knowledge, the idea of leveraging intermediate features to enhance both efficiency and performance in continual learning has not been explored before. Our work builds upon the known observations and introduces a novel approach to utilize it effectively.
>
> > The Riemer et al., 2018 paper is referred to as "ER" in the paper, which is a little bit confusing. (...)I would like the authors to confirm that the "ER" columns in the tables actually refer to the method proposed in the Riemer et al., 2018 paper.
>
> We thank the reviewer for pointing out this inaccuracy. The "ER" in our work refers to Experience Replay as described in [1], and we will update the paper to make this clear.
>
> To clarify the distinction between FT+Ex and ER, FT+Ex represents the most naive form of replay, where stored data is simply added to the training dataset for each task without modifying batching or sampling procedures. In contrast, ER ensures that each batch maintains a balanced number of samples from both old and new classes and involves oversampling from the memory. Therefore, ER is a more sophisticated replay method.
>
> >  I have some difficulty understanding what figure 4a is plotting exactly ("we plot overthinking normalized by the accuracy of the final classifier" seems a bit unclear). (...)
>
> We appreciate the Reviewer’s feedback. Given the oracle accuracy $Acc_{oracle}$ and final classifier accuracy $Acc_{final}$, normalized accuracy in Figure 4a is simply $\frac{Acc_{oracle} - Acc_{final}}{Acc_{final}}$. We will clarify this in the updated version of the paper.
>
> > In figure 2, the earlier layers (such as AC1) changes least during the process of continual learning, while AC6 changes most (closet to the final layer). However, in figure 3, we notice that AC6 provides most performance improvement for a majority of the tasks in non-trivial CL methods. Is there an intuition on why this is the case? (...)
>
> Thank you for the question. Early feature stability for older tasks is a motivation behind our approach, but it does not conflict with the better discriminative abilities of the classifiers attached to the later layers, especially on the new data. Figure 2 demonstrates classifier performance in isolation, thus later classifiers will exhibit higher downstream accuracy. Our approach leverages both of those properties to enhance overall performance in CL. Note how with different ACs we learn to classify the samples based on diverse features (e.g. more primitive features in early layers, and more sophisticated features in the later ones) and this improves the robustness of our method (as evidenced by our analysis of overthinking).
>
> We will elaborate this in more detail in the updated version of the paper.
>
> > Is there any justification on using the confidence score to choose which auxiliary classifier to use for the final prediction (...), especially given that neural networks are often not very well calibrated?
>
> Our choice of the dynamic inference aggregation rule was primarily empirical. Methods that combine multiple predictions (e.g., voting, weighted ensembling) introduce additional complexity without yielding clear improvements (see our analysis of cascading and ensembling classifiers in Appendix C.2). Therefore, we opted for the simpler approach of using a single classifier’s output. We conducted an ablation study comparing the use of the final classifier prediction versus selecting the prediction with the highest confidence (Appendix E.5), and opted to use the best-performing method. See also response to the Reviewer FvJb about thresholding strategies.
>
> **References**
>
> [1] Chaudhry et al., *"Continual learning with tiny episodic memories."*, Workshop on Multi-Task and Lifelong Reinforcement Learning (2019).

---

### Official Review · Reviewer_Y8XT · 2025-02-27

**Overall Recommendation:** 4

**Summary:**

The paper shows that intermediate representations are less prone to forgetting in the continual learning paradigm, which is aligned with previous observations in the literature. Based on this finding, they propose using auxiliary classifiers in the intermediate layers and showed that the method consistently outperforms the single classifier, when combined with various continual learning techniques.

## Update after rebuttal:
The authors have addressed my comments. I will keep my positive score.

**Claims And Evidence:**

Yes, the paper provides empirical evidence and analyses for claims. The method is combined with various continual learning techniques and evaluated on two benchmarks with different numbers of tasks. Different architectures are also analyzed.

**Essential References Not Discussed:**

Up to my knowledge, the paper covers the essential references.

**Experimental Designs Or Analyses:**

Yes. The experimental design is reasonable. One comparison that could make it stronger is the comparison against shallow architecture (with perhaps wider layers), as it is also less prone to forgetting.

**Methods And Evaluation Criteria:**

Yes, though since the method focuses on forgetting, I expect to see a metric to measure it in the experimental results. However, only the final accuracy is reported.

**Other Comments Or Suggestions:**

- CKA is mentioned in Section 1 without definition.
- Figure 2 is not clear until one reads the details in Section 3.1 (despite been discussed in Section 1).

**Other Strengths And Weaknesses:**

Strengths:
- The method is simple, yet effective
- Extensive experiments are provided
- Generalization across different architectures is analyzed
- Paper is well-written

Weakness:
- Forgetting metric is missing in the results
- Since the performance relies on earlier layers, reporting the achieved accuracy of task t at time t is useful.

**Questions For Authors:**

1. Have you compared against shallow networks?
2. Can you report the forgetting?
3. Do you think that these findings generalize to LLMs? And where to place ACs in that case?

**Relation To Broader Scientific Literature:**

Up to my knowledge, exploring the idea of early exit classifiers in the continual learning paradigm is novel, but exists in other paradigms as acknowledged in the paper.

**Theoretical Claims:**

The paper doesn’t include theoretical claims.

---

> ### Author Rebuttal · Authors · 2025-03-30
>
> We thank the Reviewer for the time spent on our work and the suggestions for improving its quality. Below, we address the points raised by the Reviewer.
>
> > Since the method focuses on forgetting, I expect to see a metric to measure it in the experimental results.
>
> We appreciate the Reviewer’s suggestion and have now included forgetting metrics for CIFAR-100. Our method generally helps in mitigating forgetting. Even in cases where we see more forgetting with ACs, it still outperforms the baseline in terms of accuracy (see Table 1).
>
> |||FT|FT+Ex|GDumb|ANCL|BiC|DER++|ER|EWC|LwF|LODE|SSIL|Avg|
> |:-|:-|:-|:-|:-|:-|:-|:-|:-|:-|:-|:-|:-|:-|
> |CIFAR100x5|Base|64.85$\pm$1.27|**44.80$\pm$0.79**|12.01$\pm$0.63|36.94$\pm$0.79|9.33$\pm$2.89|7.42$\pm$4.17|47.25$\pm$0.36|64.65$\pm$1.13|24.21$\pm$2.69|**20.61$\pm$1.14**|24.80$\pm$1.08|32.44$\pm$19.83|
> ||+AC|**50.04$\pm$1.18**|45.51$\pm$0.66|**10.00$\pm$0.47**|**26.32$\pm$1.15**|**9.26$\pm$4.00**|**3.68$\pm$4.05**|**40.31$\pm$0.84**|**48.04$\pm$1.34**|**19.95$\pm$0.57**|**20.61$\pm$0.96**|**19.95$\pm$1.12**|**26.70$\pm$15.90**|
> |CIFAR100x10|Base|70.35$\pm$2.42|49.57$\pm$1.25|14.11$\pm$1.08|40.49$\pm$1.55|9.31$\pm$2.43|11.88$\pm$4.27|51.77$\pm$1.11|68.40$\pm$2.74|**24.26$\pm$2.12**|**18.49$\pm$1.50**|21.44$\pm$2.74|34.55$\pm$21.51|
> ||+AC|**56.40$\pm$2.97**|**48.03$\pm$1.05**|**10.68$\pm$1.44**|**30.60$\pm$1.46**|**6.37$\pm$3.81**|**8.39$\pm$5.10**|**46.62$\pm$1.33**|**54.08$\pm$2.30**|29.12$\pm$1.01|20.24$\pm$2.12|**16.71$\pm$1.96**|**29.75$\pm$17.96**|
>
> Assuming we train on total of $T$ tasks, we compute forgetting as $\frac{1}{T-1} \sum_{t=1}^{T-1} (Acc_{t}^{t} - Acc_{t}^{T})$, where $Acc_{i}^{j}$ refers to the accuracy on the i-th task after training on j-th task (we skip T-th task, as it has zero forgetting according to this definition). We will add these results to the paper.
>
> > One comparison that could make it stronger is the comparison against shallow architecture (with perhaps wider layers).
>
> Thank you for the suggestion. To address this, we evaluated our method using WideResNet16 [1] on CIFAR100, which is shallower than the ResNet-32, but twice wider. Consistent with previous results, the AC-enhanced methods outperform the baselines.
>
> |||FT|FT+Ex|GDumb|ANCL|BiC|DER++|ER|EWC|LwF|LODE|SSIL|Avg|
> |:-|:-|:-|:-|:-|:-|:-|:-|:-|:-|:-|:-|:-|:-|
> |CIFAR100x5|Base|19.40$\pm$0.30|38.50$\pm$0.64|19.17$\pm$0.95|44.18$\pm$0.86|50.73$\pm$1.40|34.88$\pm$7.28|33.56$\pm$0.66|19.09$\pm$0.17|43.17$\pm$0.35|43.72$\pm$0.85|48.95$\pm$0.43|35.94$\pm$0.62|
> ||+AC|**29.79$\pm$1.94**|**40.67$\pm$0.37**|**24.09$\pm$1.16**|**46.09$\pm$0.75**|**53.38$\pm$0.68**|**46.02$\pm$1.08**|**39.61$\pm$0.64**|**32.58$\pm$0.37**|**44.26$\pm$0.02**|**50.83$\pm$0.30**|**51.35$\pm$0.53**|**41.70$\pm$0.23**|
> |CIFAR100x10|Base|9.13$\pm$0.74|34.39$\pm$0.80|21.94$\pm$0.26|33.32$\pm$0.50|46.02$\pm$0.29|34.28$\pm$3.80|30.56$\pm$0.50|10.10$\pm$0.35|32.59$\pm$1.05|40.22$\pm$0.27|45.58$\pm$0.28|30.74$\pm$0.52|
> ||+AC|**16.77$\pm$1.51**|**37.92$\pm$0.12**|**27.81$\pm$0.46**|**35.28$\pm$0.34**|**48.79$\pm$0.32**|**43.57$\pm$0.63**|**36.32$\pm$0.74**|**21.37$\pm$1.90**|**34.05$\pm$0.94**|**46.28$\pm$0.20**|**47.01$\pm$0.66**|**35.92$\pm$0.24**|
>
> > Since the performance relies on earlier layers, reporting the achieved accuracy of task t at time t is useful.
>
> Thank you for this suggestion. For the Reviewer’s convenience, we plotted such accuracies ([link](https://imgur.com/a/0O3btHy)). We will add those plots to the paper appendix.
>
> > CKA is mentioned in Section 1 without definition. Figure 2 is not clear (...).
>
> Thank you for the feedback. While CKA is a well-established metric in the literature, we acknowledge that adding a brief definition will improve clarity, and we will include it as suggested.
>
> > Do you think that these findings generalize to LLMs? And where to place ACs in that case?
>
> Prior work, such as [2], explores the usae of early exits in LLMs, suggesting that our approach could, in principle, be applicable in that context. The placement and architecture of ACs in LLMs could follow a similar strategy to what we describe for ViTs (see Appendix A).
>
> However, we are cautious about making direct claims regarding the transferability of our findings to LLMs. Our work explores class-incremental learning in vision, which differs fundamentally from language modeling, where tokenization constrains the class space. ACs might be applied to domain-incremental scenarios in LLMs, but investigating this requires further research beyond our current scope.
>
> **References**:
>
> [1] Zagoruyko et al., *“Wide Residual Networks”*, BMVC 2016
>
> [2] Schuster et al., *“Confident Adaptive Language Modeling”*, NeurIPS 2022

---

> > ### Comment · Reviewer_Y8XT · 2025-04-02
> >
> > Thank you for your response and providing extra results addressing my comments.
> >
> > I would encourage the authors to include the forgetting for imageNet as well in the revised version.
> >
> > Analyzing your new results on shallow and wide architecture, It seems that just using 'WideResNet16 base' outperforms 'base ResNet-32 + AC' in multiple cases (like Ft+Ex, ANCL, LWF, BiC) right? I am wondering what could be the motivation behind using the proposed method over shallow networks?

---

> > > ### Author Response · Authors · 2025-04-02
> > >
> > > Thank you for acknowledging our response.
> > >
> > > We will follow the Reviewer's advice to include forgetting for ImageNet in the revised version, but we need a bit more time to parse the results and re-compute some of the runs where we lost the model checkpoints. We agree that the introduction of these results should further improve our paper.
> > >
> > > Regarding shallow vs. wide architectures, we based our experimental setup on common class-incremental learning (CIL) benchmarks from the FACIL survey [1], which uses ResNet32 (0.47M parameters) for main experiments on CIFAR100 due to its balance between model size and performance. While deeper models like ResNet18 (11.2M) or VGG19 (39.3M) are sometimes used in CIL for CIFAR100, they offer no clear advantage while significantly increasing computational cost.
> > > WideResNets are less commonly used in CIL, mainly appearing in ablation studies. Upon some investigation, it seems that such wider networks are indeed overlooked in CIL - e.g. [2] argues that wider models are less prone to forgetting, and Table 1 from [3] in particular shows that, on CIFAR100, WideResNet16-2 (0.7M parameters variant we used for the results in this rebuttal) performs similarly to ResNet18 (11.2M parameters). However, our primary goal is not to optimize architecture selection but to demonstrate the general applicability of ACs in continual learning. We thank the Reviewer for providing an insightful suggestion for another evaluation case for our idea, which further strengthens our paper.
> > >
> > > **References**
> > >
> > > [1] Masana et al., *"Class-incremental learning: survey and performance evaluation on image classification"*, TPAMI 2022
> > >
> > > [2] Mirzadeh, et al. *"Wide neural networks forget less catastrophically"*, ICML 2022
> > >
> > > [3] Mirzadeh et al., "*Architecture Matters in Continual Learning"*, 2022
> > >
> > > **(edit) Updated partial forgetting data on ImageNet100 with ResNet18**
> > >
> > > For the Reviewer's sake, we present forgetting data for ImageNet100 runs for runs computed so far. We will update the paper with the full table containing all the methods, but were unable to do so in short rebuttal period due to the computational constraints.
> > >
> > > |Setting|Setup|FT|FT+Ex|GDumb|BiC|ER|EWC|LwF|SSIL|Avg|
> > > |:-|:-|:-|:-|:-|:-|:-|:-|:-|:-|:-|
> > > |ImageNet100x5|Base|74.35$\pm$0.63|52.00$\pm$0.75|16.73$\pm$0.82|10.37$\pm$1.70|56.14$\pm$1.27|74.49$\pm$0.97|18.90$\pm$0.73|21.90$\pm$0.88|40.61$\pm$24.91|
> > > ||+AC|61.45$\pm$1.30|49.97$\pm$1.00|15.98$\pm$1.07|9.76$\pm$1.05|49.65$\pm$1.25|61.77$\pm$0.61|17.49$\pm$0.76|19.68$\pm$0.80|35.72$\pm$20.59|
> > > |ImageNet100x10|Base|77.50$\pm$1.54|57.50$\pm$1.40|19.67$\pm$1.37|12.73$\pm$3.38|60.38$\pm$1.42|76.70$\pm$1.40|23.76$\pm$1.49|19.75$\pm$1.69|43.50$\pm$25.51|
> > > ||+AC|68.44$\pm$1.54|54.67$\pm$1.73|18.12$\pm$1.47|12.84$\pm$2.71|56.06$\pm$1.74|68.13$\pm$1.18|18.17$\pm$0.90|18.90$\pm$1.80|39.42$\pm$22.94|

---

### Official Review · Reviewer_dDEJ · 2025-03-06

**Overall Recommendation:** 3

**Summary:**

The manuscript analyzes the effect of class-incremental training on the parameters of a neural network, finding that deeper layers tend to be more affected from catastrophic forgetting. To exploit this it introduces a series of Auxiliary Classifiers (AC), finding that their combination can significantly make the predictions more robust.

**Claims And Evidence:**

The claims are convincing.

**Essential References Not Discussed:**

N/A

**Experimental Designs Or Analyses:**

The choice of benchmarks and competitors is adequate. However, I found the use of a single seeded run for the results of the ViT-based models (lines 301-302) to be concerning. In my opinion, for a proper evaluation the tests should be repeated across multiple runs without seeds.

**Methods And Evaluation Criteria:**

The choice of benchmarks and competitors seems adequate.

**Other Comments Or Suggestions:**

N/A

**Other Strengths And Weaknesses:**

- While the manuscript investigates the change in the intermediate representations for a backbone trained from scratch, it could be interesting to analyze the effect of AC on pre-trained backbones, especially in light of [1] which conducts the same initial tests with CKA for pre-trained networks.
- How much is the overhead of computing all gradients for the losses of the ACs (e.g., in terms of required GPU memory and training time with respect to the baseline)?

[1]: Boschini et al. "Transfer without forgetting." ECCV 2022.

**Questions For Authors:**

See section on "Other Strengths And Weaknesses"

**Relation To Broader Scientific Literature:**

While the analysis on the change in the intermediate representations in a class-incremental environment has been conducted in several other works [1,2,3] (with [1,2] being already cited in the manuscript), the use of auxiliary classifiers to take advantage of the smaller changes in parameters of shallower layers seems novel to me.

[1]: Ramasesh, V. V. et al. “Anatomy of catastrophic forgetting: Hidden representations and task semantics.” In ICLR 2020.
[2]: Zhao, Haiyan et al. “Does continual learning equally forget all parameters?." ICML 2023.
[3]: Boschini et al. "Transfer without forgetting." ECCV 2022.

**Theoretical Claims:**

N/A

---

> ### Author Rebuttal · Authors · 2025-03-30
>
> We thank the Reviewer for the time spent on our work and the suggestions for improving its quality. We appreciate the Reviewer’s mention of *Boschini et al., "Transfer without forgetting." ECCV 2022*;  we will include this paper in the updated related works. Below, we address the points raised by the Reviewer.
>
> > I found the use of a single seeded run for the results of the ViT-based models (lines 301-302) to be concerning. In my opinion, for a proper evaluation the tests should be repeated across multiple runs without seeds.
>
> We appreciate the Reviewer’s concern. Due to computational constraints, we initially conducted these experiments with a single seed, as ViT-based models are particularly expensive to train. In response to the Reviewer’s suggestion, we have now rerun the ViT experiments from Figure 7 using two additional seeds (three in total). The updated figure, including confidence intervals, is available at [link](https://imgur.com/JTuI5wn), demonstrating that our findings remain consistent across multiple runs. We will incorporate these results into the revised version of our paper.
>
> > While the manuscript investigates the change in the intermediate representations for a backbone trained from scratch, it could be interesting to analyze the effect of AC on pre-trained backbones, especially in light of [1] which conducts the same initial tests with CKA for pre-trained networks.
>
> Thank you for the interesting suggestion. We conducted warm-start continual learning experiments (Appendix B.1) as a controlled alternative to pretraining, empirically demonstrating that our approach remains effective in such a setting. We consider the investigation of pre-trained models with ACs a promising future work direction.
>
> > How much is the overhead of computing all gradients for the losses of the ACs (e.g., in terms of required GPU memory and training time with respect to the baseline)?
>
> We appreciate the Reviewer’s question. We measured training times and peak memory usage for training ResNet32 with different numbers of ACs on CIFAR-100, summarized in the tables below. "0 ACs" refers to a standard network. We report training times in hours (mean from three runs) and peak memory usage in GB. For memory, we include one table, as it does not depend on the data split.
>
> ## Training times (in hours) - 5 tasks
> |ACs|ANCL|BiC|ER|EWC|FT|FT+Ex|GDUMB|LODE|LwF|SSIL|Avg|
> |-:|-:|-:|-:|-:|-:|-:|-:|-:|-:|-:|-:|
> |0|2.1|1.4|2.4|1.2|1.1|1.3|0.6|3|1.2|1.7|1.6|
> |3|2.7|1.8|2.8|1.5|1.3|1.5|0.8|3.6|1.5|2.1|2|
> |6|3.3|2.1|3.3|1.8|1.6|1.7|1.1|4.3|1.8|2.5|2.4|
> |12|4.5|2.9|4|2.5|2|2.3|1.5|5.5|2.5|3.2|3.1|
>
> ## Training times (in hours) - 10 tasks
> |ACs|ANCL|BiC|ER|EWC|FT|FT+Ex|GDUMB|LODE|LwF|SSIL|Avg|
> |-:|-:|-:|-:|-:|-:|-:|-:|-:|-:|-:|-:|
> |0|2.8|2|2.5|1.4|1.4|1.7|1.2|3.3|1.4|1.8|1.9|
> |3|3.6|2.5|3.2|1.9|1.6|2|1.5|4.2|1.9|2.3|2.5|
> |6|4.4|3|3.8|2.2|1.9|2.4|1.7|5.2|2.4|2.9|3|
> |12|6.8|4.2|4.8|3|2.4|3.3|2.3|6.9|3.5|3.9|4.1|
>
> # Peak GPU memory usage (GB)
> |ACs|ANCL|BiC|ER|EWC|FT|FT+Ex|GDUMB|LODE|LwF|SSIL|Avg|
> |-:|-:|-:|-:|-:|-:|-:|-:|-:|-:|-:|-:|
> |0|2.1|2.09|2.1|2.1|2.1|2.1|2.39|2.1|2.1|2.18|2.14|
> |3|2.29|2.22|2.18|2.56|2.25|2.18|2.56|2.22|2.22|2.29|2.3|
> |6|2.47|2.32|2.27|2.73|2.41|2.27|2.73|2.31|2.3|2.41|2.42|
> |12|2.82|2.5|2.43|3.02|2.72|2.43|3.03|2.49|2.5|2.65|2.66|
>
> Our standard setup (6 ACs) results in approximately 50% training time and 10% memory overhead. The exact overheads will depend on the model, AC architecture, and training environment. However, we do not consider this overhead prohibitive for real-world use cases, particularly in offline class-incremental learning, where training resource constraints are not usually strict; therefore, we also did not optimize the code towards such constraints. As our primary concerns are efficiency and performance during the inference time, we consider the discussed overheads acceptable. We appreciate the Reviewer raising this important point, and we will incorporate this discussion in the paper.

---

### Official Review · Reviewer_FvJb · 2025-03-21

**Overall Recommendation:** 3

**Summary:**

This paper introduces Auxiliary Classifiers (ACs) to enhance performance and efficiency in Continual Learning (CL) by leveraging intermediate representations in neural networks. The primary challenge is catastrophic forgetting, where new knowledge acquisition disrupts previously learned information. The key conceptual idea is that intermediate neural network representations exhibit higher stability than final-layer representations, making them less prone to forgetting. The authors propose attaching auxiliary classifiers (ACs) to intermediate layers to Improve accuracy by reducing overfitting in later layers, accelerate inference through early classification when confidence is high, and enhance generalization by leveraging more stable feature representations. The paper evaluates ACs across multiple continual learning methods using CIFAR100 and ImageNet100 datasets.

**Claims And Evidence:**

* The paper is mostly clearly written, making it easy to follow the authors' claims and methodologies. However,
* Figure 3 is unclear. It appears that all auxiliary classifiers outperform the final classifier (non-negative accuracy differences). If this is not the intended message, the figure needs clarification or revision.
* Figure 5 is not clearly presentated. Specifically, what do the check marks and x-marks represent? Clarification is also needed on how this figure illustrates static inference as described in the accompanying paragraph.

**Essential References Not Discussed:**

No essential missing references were identified in the current version of the paper.

**Experimental Designs Or Analyses:**

* The experimental design is sound and comprehensive, covering several baseline comparisons and tasks relevant to continual learning.
* However, clarity regarding the training procedures of auxiliary classifiers (e.g., incremental class additions, gradient propagation handling) needs improvement. For example, how are new classes incrementally added for each AC? Are ACs initialized with a fixed number of classes, or do classes expand progressively? Additionally, the authors mention "enabled gradient propagation" in Section 3.2, but it is unclear whether this explicitly means gradients are not detached. If this is the case, could the authors clarify the reasoning behind this choice? Moreover, could gradient propagation negatively affect intermediate representations, potentially causing performance degradation, as suggested by the LwF case in Figure 4(d)?

**Methods And Evaluation Criteria:**

* The experimental design is thorough, valid, and effectively covers relevant scenarios.
* Continual Learning literature commonly reports metrics such as average accuracy and average backward transfer (BWT). However, this paper primarily focuses on final-task accuracy. I believe Tables 1 and 2, and Figures 1 and 6, should includ average accuracy and BWT.
* The authors provide inference costs, but training costs should also be reported, given that adding auxiliary classifiers, both with and without gradient detachment, could significantly impact computational overhead.

**Other Comments Or Suggestions:**

* Figures and their corresponding textual explanations often appear on separate pages, negatively impacting readability. Repositioning figures closer to their descriptions would greatly enhance readability.
* For figure 1 and 6, it should be stated if the accuracy is average task accuracy or final task accuracy.
* Including explicit accuracy values for each AC in Figure 4 would be helpful, as parts (b) and (c) currently show only relative performance.
* Defining overthinking as "cases where samples correctly classified by early classifiers are misclassified by the final classifier" is intuitive but may oversimplify the phenomenon. A more nuanced justification or further context would strengthen this definition.

**Other Strengths And Weaknesses:**

None

**Questions For Authors:**

* Do ACs also suffer catastrophic forgetting, trained with and without graident detach? If ACs are continually trained alongside the final classifier, it seems plausible that ACs would also forget older tasks. Could you discuss or demonstrate the robustness of ACs across sequential tasks?
* While the paper demonstrates robustness in early representations, it also acknowledges later layers’ superior ability to classify larger subsets. What do you believe explains this phenomenon?
* In Figure 4, "unique overthinking" appears relatively minor (around 10%). Could the authors elaborate further on the significance or implications of this proportion?
* If confidence measures play a key role in your inference strategy, it would be valuable to analyze and report confidence metrics explicitly for each AC.
* A common viewpoint in deep learning literature suggests that early layers learn general representations, while later layers acquire task-specific representations. While authors claim that early layers do not "forget old tasks," it could be also possible that the apparent stability of early layers might reflect learned shared characteristics (e.g., color, shape) rather than true resistance to forgetting. What is your perspective on this interpretation?

**Relation To Broader Scientific Literature:**

The paper distinguishes the novel usage of intermediate auxiliary classifiers from existing CL techniques.

**Theoretical Claims:**

* The paper does not present explicit theoretical claims or proofs.

---

> ### Author Rebuttal · Authors · 2025-03-30
>
> We thank the Reviewer for all the suggestions. Below, we address the points mentioned by the Reviewer. We will include the clarifications and the Reviewer’s suggestions in the updated version of our paper.
>
> ## Accuracy metrics and backward transfer (BWT)
>
> In all our experiments, we report the average accuracy across all classes after the final task.
>
> BWT can be seen as the negative of forgetting, which we discussed in the response to the Reviewer Y8XT. It is hard to observe BWT consistently in our setting across the whole training course and interpret this metric, so we do not report it like most class-incremental learning works (the metric is mostly used in task-incremental settings). However, BWT can be derived from the per-task accuracy plots we added in response to Reviewer Y8XT ([link](https://imgur.com/a/0O3btHy)).
>
> ## Early vs later layer representation stability
>
> We agree with the reviewer that feature-sharing is a useful concept when explaining the results observed in our paper. Early-layer representations tend to capture shared, low-level features that are less task-specific, making them more stable across tasks and hence suffer less from forgetting. The more complex, later-layer features, are more task-specific and consequently suffer from more forgetting (since they might not be required for current task data). As a consequence of this phenomenon, during continual learning the ACs based on lower layers can potentially outperform those of the later layers.
>
> ## Forgetting in the ACs
>
> As noted by the Reviewer, our ACs are still classifiers and as such are also susceptible to forgetting. Our approach leverages the stability of earlier features shown in Figure 2 but also benefits from the diversity of features used by different ACs. Since we classify based on the information aggregated across multiple ACs, it is enough if just one classifier returns a confident prediction for a given sample to make the correct prediction. Empirical results indicate that AC-based networks generally exhibit lower forgetting on a per-task basis across continual learning (see the forgetting plots under the [link](https://imgur.com/a/0O3btHy) and the response to Reviewer Y8XT), supporting our intuition that using multiple ACs results in more robust classification.
>
> ## Thresholding strategy
>
> The Reviewer correctly points out that our inference strategy is influenced by prediction confidence, which varies for each AC and the task. While confidence-aware inference strategies could improve performance, they are complex enough to warrant separate papers (e.g., Meronen et al., *“Fixing Overconfidence in Dynamic Neural Networks”*, WACV 2024) and often rely on calibration on a holdout dataset, which limits their applicability to exemplar-based methods. To maintain simplicity and generalizability, we use a shared threshold. The fact that our approach performs well without per-AC adaptation further highlights its robustness and broad applicability.
>
> ## Overthinking definition
>
> Our use of the term “overthinking” follows the SDN paper and is well-established in the literature, so we skipped it to keep the paper concise. Figures 4b and 4c are supposed to provide a more nuanced analysis of the phenomenon.
>
> ## Unique overthinking
>
> "Unique overthinking" indicates how many samples can be correctly classified by a unique AC. It highlights that each classifier specializes in different subsets of the data in a non-redundant way.
>
> ## How are the ACs initialized?
>
> We follow the same FACIL protocol for both the main classifier and the auxiliary classifiers (ACs) - we add a new head for the classes introduced in each task.
>
> ## Gradient propagation in Section 3
>
> We mention that gradients from ACs are detached from the backbone in Section 3.2 (L#201-204); in Section 3.4, we compare this with a setup where gradient propagation is enabled end-to-end to empirically evaluate which approach yields better results. Then in all following experiments, we used the setup with enabled gradient propagation.
>
> Gradient propagation can sometimes negatively affect individual classifier performance. However, while some individual classifiers may degrade slightly, the overall benefit outweighs these effects. In most cases, gradient propagation training leads to better results (see Appendix C.3.). In Appendix E, we also show that findings from our analysis are consistent across with both detached and enabled gradients.
>
> ## Training times with ACs
>
> See the response to the Reviewer dDEJ.
>
> ## Figures clarification
>
> Figure 3 is intended to highlight which auxiliary classifiers outperform the final classifier.
>
> The marks in Figure 5 are supposed to represent correct and incorrect predictions.
>
> The accuracy reported in Figures 1 and 6 is the average accuracy after the final task.

---

### Decision · Program_Chairs · 2025-05-01

**Decision:**

Accept (poster)

**Comment:**

The paper found that intermediate layers' representation is less prone to forgetting, leading to better performance. The paper empirically shows that the intermediate layers' representation is effective for less forgetting by proposing an auxiliary classifiers that use the intermediate layers' representations. All reviewers appreciate the benefit of the proposed method. Most of reviewers discuss with the authors with the rebuttal. Agreeing to the reviewers' suggestion, the AC recommends to accept the paper to ICML 2025.